# Translation is required for miRNA-dependent decay of endogenous transcripts

Adriano Biasini[1] (iD), Baroj Abdulkarim[1], Stefano de Pretis[2], Jennifer Y Tan[1] (iD), Rajika Arora[3], Harry Wischnewski[3], Rene Dreos[4], Mattia Pelizzola[2] (iD), Constance Ciaudo[3] (iD) & Ana Claudia Marques[1,*] (iD)

## Abstract

Post-transcriptional repression of gene expression by miRNAs occurs through transcript destabilization or translation inhibition. mRNA decay is known to account for most miRNA-dependent repression. However, because transcript decay occurs co-translationally, whether target translation is a requirement for miRNA-dependent transcript destabilization remains unknown. To decouple these two molecular processes, we used cytosolic long noncoding RNAs (lncRNAs) as models for endogenous transcripts that are not translated. We show that, despite interacting with the miRNA-loaded RNA-induced silencing complex, the steady-state abundance and decay rates of these transcripts are minimally affected by miRNA loss. To further validate the apparent requirement of translation for miRNA-dependent decay, we fused two lncRNA candidates to the 3'-end of a protein-coding gene reporter and found this results in their miRNA-dependent destabilization. Further analysis revealed that the few natural lncRNAs whose levels are regulated by miRNAs in mESCs tend to associate with translating ribosomes, and possibly represent misannotated micropeptides, further substantiating the necessity of target translation for miRNA-dependent transcript decay. In summary, our analyses suggest that translation is required for miRNA-dependent transcript destabilization, and demonstrate that the levels of coding and noncoding transcripts are differently affected by miRNAs.

**Keywords** Dicer knockout mESC; long noncoding RNAs; miRNA; RNA metabolic labelling; translation

**Subject Categories** RNA Biology; Translation & Protein Quality

**The EMBO Journal (2021) 40: e104569**

## Introduction

Post-transcriptional regulation of gene expression by microRNAs (miRNAs) is widespread in eukaryotes and impacts diverse biological processes in health and disease (Bartel, 2004; Bartel, 2009). Most mature miRNAs are the product of a relatively complex biogenesis process. Primary miRNA transcripts that generally depend on RNA Polymerase II for transcription are initially processed by the nuclear enzyme DROSHA and its cofactor DGCR8 into a premature hairpin RNA ~60 nucleotides in length (pre-miRNA transcript; Lee *et al*, 2003). Pre-miRNAs are exported into the cytoplasm, where they undergo a second round of processing by DICER, resulting in a ~22 nucleotide long double-stranded RNA duplex (Hutvagner *et al*, 2001). Loss-of-function mutations in any of the miRNA-processing factors result in complete depletion of most miRNA species (Kim *et al*, 2016). Argonaute proteins (AGO) bind mature miRNAs and guide target recognition of the RNA-inducing silencing complex (RISC). In mammals, target recognition relies primarily on complementarity between the miRNA seed region (position 2–8 of the mature miRNA) and miRNA recognition elements (MREs) in the target (Bartel, 2018).

Post-transcriptional repression by miRNAs occurs by translation inhibition or transcript decay (Bartel, 2009). The relative contributions of RNA destabilization and translation inhibition to miRNA-dependent repression have been extensively studied (Iwakawa & Tomari, 2015; Jonas & Izaurralde, 2015). These studies support the general consensus that translation inhibition precedes transcript deadenylation and decay (Baek *et al*, 2008; Selbach *et al*, 2008; Bethune *et al*, 2012), which in turn is thought to account for most miRNA-dependent repression (Baek *et al*, 2008; Selbach *et al*, 2008; Pitchiaya *et al*, 2012). The coupling between translation inhibition and transcript destabilization is further substantiated by evidence that protein-coding transcripts undergoing miRNA-dependent repression are associated with translating ribosomes (Wightman *et al*, 1993; Olsen & Ambros, 1999; Seggerson *et al*, 2002; Nottrott *et al*, 2006; Petersen *et al*, 2006; Gu *et al*, 2009; Tat *et al*, 2016) and that most miRNAs loaded into RISC (miRISC) co-localize with polysomes (Kim *et al*, 2004; Nelson *et al*, 2004; Maroney *et al*, 2006).

These observations have raised questions regarding the requirement of translation for miRNA-dependent transcript decay. A number of experiments relying on the analysis of reporter constructs

1 Department of Computational Biology, University of Lausanne, Lausanne, Switzerland
2 Center for Genomic Sciences, Istituto Italiano di Tecnologia (IIT), Milano, Italy
3 Institute of Molecular Health Sciences, ETHZ, Zurich, Switzerland
4 Center for Integrative Genomics, University of Lausanne, Lausanne, Switzerland
*Corresponding author. E-mail: anaclaudia.marques@unil.ch

revealed that transcript decay occurs even when translation initiation or elongation are impaired (Wakiyama *et al*, 2007; Eulalio *et al*, 2009; Fabian *et al*, 2009). However, it is hard to reconcile the extent of target repression reported in these studies (up to 10-fold) with the well-established impact of most miRNAs on endogenous transcript abundance, which rarely exceeds twofold (Baek *et al*, 2008; Selbach *et al*, 2008). This has prompted concerns on whether exogenously expressed reporters faithfully recapitulate the behaviour of the majority of endogenously expressed transcripts.

To assess the requirement of translation for miRNA-dependent destabilization of endogenous targets, and to overcome some of the limitations that may arise from using exogenous reporters, we took advantage of endogenously expressed cytosolic intergenic long noncoding RNAs (lncRNAs). This class of noncoding transcripts rarely associates with ribosomes (Guttman *et al*, 2013) and has previously been shown to interact with the miRISC machinery (Helwak *et al*, 2013). These transcripts thus provide a unique opportunity to address the outstanding question of whether miRNA-dependent decay occurs in the absence of translation. Specifically, we used 4-thio-uridine (4sU) to assess genome-wide decay rates in wild-type (WT) and miRNA-depleted cells. Our genome-wide analysis revealed that the decay rates of protein-coding miRNA targets are significantly reduced upon miRNA loss, whereas those of lncRNAs are only minimally affected. Putative micropeptides were enriched among lncRNAs responsive to changes in miRNA abundance, suggesting that translation is required for miRNA-dependent decay. We validated this hypothesis experimentally by inducing association of candidate lncRNAs with translating ribosomes, and found that this is sufficient to induce miRNA-dependent decay, further substantiating the prerequisite of translation for miRNA-dependent transcript destabilization.

# Results

## Cytosolic lncRNAs interact with miRISC

Since post-transcriptional regulation by miRNAs occurs in the cytoplasm (Bartel, 2018), and does not directly impact the levels of nuclear lncRNAs, we first classified lncRNAs based on their subcellular localization. We used RNA sequencing data from mESCs nuclear and cytosolic fractions (Tan *et al*, 2015) to estimate the expression of protein-coding transcripts (mRNAs) and intergenic long noncoding RNAs (lncRNAs) in these two subcellular compartments (Fig EV1A). We considered lncRNAs with a cytoplasmic/nuclear expression ratio higher than the median ratio for mRNAs, which are predominantly cytoplasmic, highly to be cytosolic ($n = 1,081$). The remaining mESC lncRNAs were considered to be nuclear ($n = 4,953$). Ribosome profiling data in mESCs (Ingolia *et al*, 2011) support that mRNAs (50.4%) are more frequently associated with translating ribosomes than cytosolic or nuclear lncRNAs (6.6% and 4.0%, respectively, two-tailed chi-square test, $P$-value $< 2 \times 10^{-16}$, Fig 1A). We took advantage of publicly available Halo-enhanced Ago2 pull-down (HEAP) data in mESCs (Li *et al*, 2020) to assess whether cytosolic lncRNAs are associated with miRISC. We found that the fraction of mESC-expressed mRNAs with experimental evidence for AGO2 binding is significantly higher than for cytosolic lncRNAs (25.7% and 4.7%, respectively, two-tailed chi-square test, $P$-value $< 2 \times 10^{-26}$). Our ability to detect binding

by miRISC using this approach is in part limited by the endogenous transcript expression, as highlighted by the significantly higher expression of transcripts bound by AGO2 (average expression (TPM) bound = 5.3 vs unbound = 0.49, two-tailed Mann–Whitney *U*-test, $P < 2 \times 10^{-26}$, Fig EV1B). Since lncRNAs are in general less expressed than mRNAs, the proportion of lncRNAs bound by AGO2 may be higher than what is detected. Despite this limitation, the density of binding at cytosolic lncRNAs (1.2 sites per kb of sequence) and mRNA 3'UTRs (1.4 sites per kb of sequence for the transcripts with experimental evidence of AGO2 binding (>0 peak)) is similar (two-tailed Mann–Whitney *U*-test, *P*-value = 0.166, Fig 1B). As expected, cytosolic lncRNAs have a significantly higher (two-tailed Mann–Whitney *U*-test, *P*-value = 0.002) density of AGO2 peaks than nuclear lncRNAs (0.7 sites per kb). Using publicly available AGO2-CLIP data in wild-type and DICER knockout mESCs (Leung *et al*, 2011), we validated the observations that the fraction and density of AGO2 binding is similar for cytosolic lncRNAs and mRNA 3'UTRs (Fig EV1C). The fraction of cytosolic lncRNAs bound by AGO2 (Leung *et al*, 2011), with (6%) and without (7%) experimental evidence of ribosomal association, is statistically indistinguishable (two-tailed Fisher's exact test $P = 0.826$), suggesting that AGO2 binding is independent of translation. We conclude that, consistent with previous analysis, most cytosolic lncRNAs do not stably associate with translating ribosomes (Guttman *et al*, 2013), but are nevertheless targeted by miRISC (Helwak *et al*, 2013), and are therefore uniquely suitable to assess miRNA-dependent destabilization of endogenous transcripts in the absence of translation.

## Steady-state expression of noncoding transcripts is minimally impacted by miRNAs

We first sought to determine whether cytosolic lncRNA expression was post-transcriptionally regulated by miRNAs. We took advantage of an mESC line containing two Cre/LoxP sites flanking the *Dicer* RNAse III domain on exon 21, and a *Cre* recombinase gene expressed under the control of a 4-hydroxytamoxifen(4-OHT)-inducible promoter (Cobb *et al*, 2005; Nesterova *et al*, 2008). Exposure of these cells to 4-OHT leads to LoxP site recombination and concomitant depletion of DICER (two-tailed paired *t*-test *P*-value = 0.010, Fig EV1D and E). Conditional loss of DICER function minimally impacts cell proliferation (two-tailed paired *t*-test *P*-value = 0.011, Fig EV1F), and the transcript and protein levels (Fig EV1G and I) of the pluripotency-associated transcription factors *Nanog*, *Oct4* and *Sox2*. In contrast to what was previously reported for *Dicer* constitutive knockdown mESCs that exhibit a 10-fold downregulation of *c-Myc* expression (Zheng *et al*, 2014), in conditional *Dicer* mESC mutants, the expression of this gene is only minimally impacted (two-tailed paired *t*-test *P*-value = 0.008, fold change between KO and WT < 0.5, Fig EV1G), supporting that this system is better suited to investigate the direct effects of miRNA depletion.

We profiled small RNA expression following DICER loss of function and found that 8 days after 4-OHT addition, mature miRNA levels are reduced by ~80% (Fig 1C). We validated these results by TaqMan RT–qPCR, for miR-295-3p and miR-290-3p, which are among the most abundant miRNAs in mESCs (Judson *et al*, 2009; Leung *et al*, 2011; Fig EV1J). Decreased levels of these miRNAs is associated, as expected, with a significant increase in the levels of some of their well-established targets (Wang *et al*, 2008; Fig EV1K).

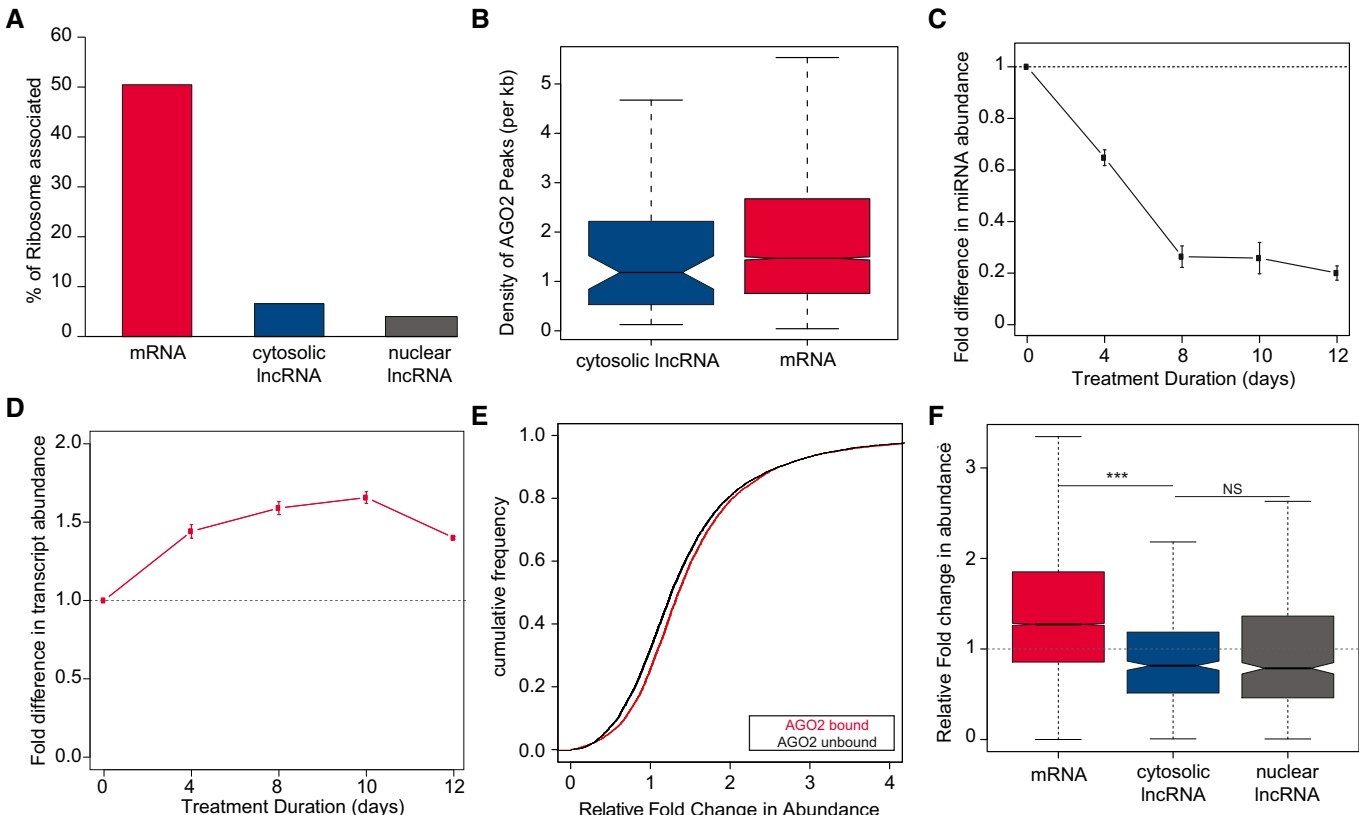

**Figure 1. Steady-state abundance of lncRNAs is not directly affected by miRNA loss.**

A   Percentage of mRNAs ($n$ = 6,701, red) and predominantly cytosolic ($n$ = 57, blue) and nuclear lncRNAs ($n$ = 175, grey) with experimental evidence of binding by ribosomes (Translation Efficiency > 0) in mESCs.

B   Density of HEAP-AGO2 peaks across cytosolic lncRNAs ($n$ = 62, blue) and the 3'UTR regions of mRNAs ($n$ = 8,798, red) with experimental evidence for AGO2 binding in mESCs (>0 AGO2 peaks). Central band of boxplot represents median, box depicts 25–75 quantiles of distribution, and whiskers represent the $5^{th}$ and $95^{th}$ quantiles of the distribution.

C, D   Small RNA and Poly(A)-selected RNA sequencing based estimates of the fold difference ($y$-axis) in (C) miRNA and (D) mRNA expression, respectively, relative to day 0, during a 12 days' time course ($x$-axis) following treatment of DTCM23/49XY mESCs with 4-OHT and consequent loss of DICER function. Points represent the average miRNA or mRNA expression and error bars the standard deviation based on three independent biological replicates.

E   Cumulative distribution plot of the fold difference in expression after 8 days of 4-OHT treatment for mRNAs, relative to day 0 of treatment (tpm ≥ 1) with ($n$ = 6,034) and without ($n$ = 7,887) evidence of AGO2-binding by HEAP (Li et al, 2020).

F   Distribution of the relative fold change after 8 days of 4-OHT treatment in steady-state abundance, relative to day 0 of treatment, for mESC-expressed (tpm ≥ 1) mRNAs ($n$ = 19,306, red), cytosolic ($n$ = 445, blue) and nuclear ($n$ = 529, grey) lncRNAs (two-tailed Mann–Whitney $U$-test, mRNAs vs cytosolic lncRNAs $P$-value < $2 \times 10^{-16}$ and cytosolic vs nuclear lncRNAs $P$ = 0.875). Statistics: NS-$P$-value > 0.05 and ***$P$-value < 0.001. Central band of boxplot represents median, box depicts 25–75 quantiles of distribution, and whiskers represent the $5^{th}$ and $95^{th}$ quantiles of the distribution. Two-tailed Mann–Whitney $U$-test.

To assess the genome-wide impact of miRNA loss on mRNA and lncRNA expression, we used data from our previously published transcriptome-wide expression profiling, following loss of DICER, experiment in these cells (Tan *et al*, 2015). As expected, and consistent with the role of miRNAs on post-transcriptional repression of protein-coding gene expression, we found that mRNA levels increased moderately, but significantly, following *Dicer* loss of function (Fig 1D). The fold increase in expression, relative to control, in miRNA-depleted mESCs is significantly higher (two-tailed Mann–Whitney $U$-test, $P = 1 \times 10^{-10}$) for transcripts with experimental evidence for AGO2 binding (Fig 1E), supporting that the observed changes in mRNA expression are, at least in part, a consequence of mRNA alleviation from miRNA-mediated repression. In contrast to mRNAs, we found that lncRNA expression was minimally impacted by miRNA depletion (Fig 1F). Specifically, and in contrast to

mRNAs, lncRNAs steady-state abundance is slightly decreased in miRNA-depleted cells (Fig 1F). This small decrease is likely an indirect effect of miRNA loss, because decreased levels of miRNAs are expected to result in increased steady-state abundance of targets, as observed for mRNAs (Fig 1F). Furthermore, the impact of miRNA depletion is similar on both subcellular classes of lncRNA and independent of co-localization with miRISC (Fig 1F). We conclude that despite interacting with miRISC, cytosolic lncRNA transcript levels are not directly regulated by miRNAs (Fig 1F).

## No evidence for miRNA-dependent destabilization of noncoding transcripts

Of the three processes (transcription, processing and degradation) that determine steady-state transcript abundance, only the rate of

degradation is directly influenced by miRNAs. To determine transcriptome-wide differences in degradation rate between miRNA-depleted and wild-type mESCs, we performed, in duplicate, 4-thio-uridine (4sU, 200 μM) metabolic labelling of RNA for 10 and 15 min, as previously described (Biasini & Marques, 2020) 8 days after induction of DICER loss of function, as well as in uninduced control mESCs. We sequenced pre-existing and newly synthesized

RNA and quantified intron and exon expression transcriptome-wide in both RNA fractions (Fig 2A, Methods). Principal component analysis revealed that the RNA fraction (pre-existing or newly synthetized) is the strongest contributor to differences in gene expression between samples (Fig EV2A). The second principal component is strongly correlated with miRNA content in the cell as a result of DICER presence or absence (Fig EV2B). Finally, the third principal

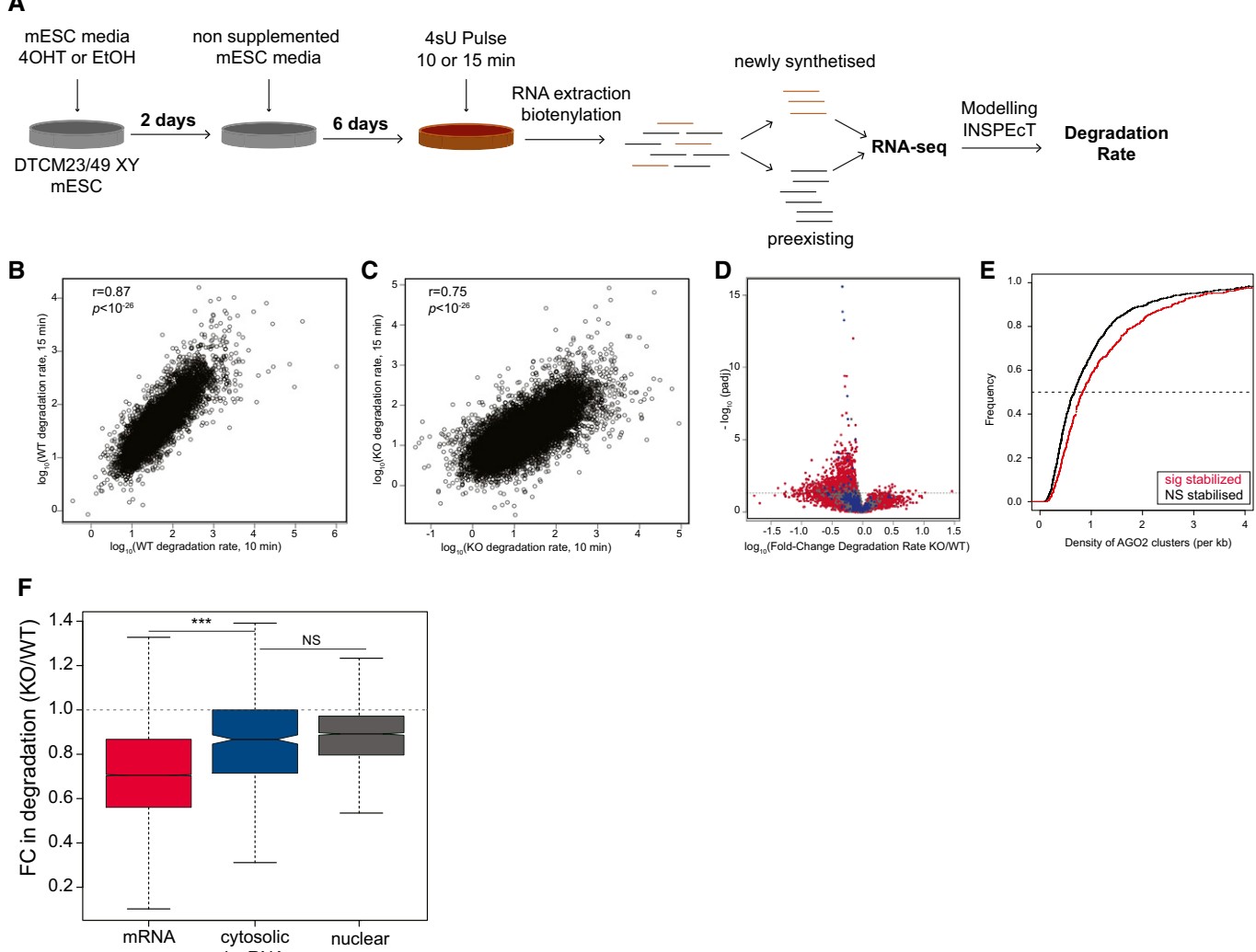

**Figure 2. No evidence for miRNA-dependent destabilization of cytosolic lncRNAs.**

A    Schematic representing 4sU metabolic labelling of conditional *Dicer* knockout and wild-type cells experiment.

B, C    Correlation (Pearson) between degradation rates ($\log_{10}$) obtained after 10 (*x*-axis) and 15 (*y*-axis) minutes of 200 μM 4sU labelling in wild-type (WT) and DICER null (KO) cells.

D    Volcano plot showing the adjusted *P*-value (*y*-axis) as a function of the fold change in degradation rate estimates, based on the 10 min pulse, between KO and WT cells (*x*-axis) for protein-coding genes (red), cytosolic (blue) and nuclear (grey) lncRNAs. Each point represents a transcript and the horizontal dashed line represents the significance cut-off.

E    Cumulative distribution plot of the density of AGO2 clusters in the 3'unstralated regions of AGO2 bound mRNAs (AGO2 cluster > 0) whose degradation rates were either significantly (*n* = 711, red) or not significantly changed (*n* = 1,127, black) between KO and WT cells, based on the 10 min pulse estimates. Density of clusters presented in this analysis was estimated based on data from (Leung *et al*, 2011).

F    Distribution of the fold change (FC) in degradation rates of mRNAs (*n* = 29,900, red), cytosolic (*n* = 474, blue) and nuclear (*n* = 2,348, grey) lncRNAs, in 4-OHT-treated (KO) relative to ethanol-treated (WT) cells after 8 days of treatment (estimated based on the 10 min 4sU pulse), horizontal dashed line represents a KO/WT FC in degradation rate of 1 Statistics: NS-*P*-value > 0.05 an d ***P*-value < 0.001. Central band of boxplot represents median, box depicts 25–75 quantiles of distribution, and whiskers represent the 5th and 95th quantiles of the distribution (two-tailed Mann–Whitney *U*-test).

component discriminates samples from different biological replicates (Fig EV2C). We estimated degradation rates from the data we obtained from the two pulse durations (10 and 15 min) separately using INSPEcT ((de Pretis *et al*, 2015), Methods). We found that these rates are highly correlated between the two pulse lengths for wild-type and miRNA-depleted cells ($R^2 > 0.75$, Fig 2B and C). As a control, we used an alternative method (transcription block by actinomycin-D) to validate the estimated differences in transcript stability between wild-type and miRNA-depleted cells for a subset of transcripts spanning a range of fold differences in degradation rates (Pearson $R^2 = 0.58$, Fig EV2D).

As expected, miRNA depletion is associated with an increase in transcript stability (85% and 91% of transcripts have lower degradation rate in KO cells for 10 and 15 min pulse, Figs 2D and EV2E, respectively). Next, we identified genes whose degradation rate is significantly different (FDR < 0.05) between miRNA-depleted and control mESCs (14% and 17% in 10 and 15 min pulse, Figs 2D and EV2E, respectively) and found that, as expected, mRNAs are significantly more often stabilized in miRNA-depleted mESCs relative to control. Finally, and consistent with a role of miRNAs in controlling the observed differences in degradation rates, transcripts whose decay rates are significantly decreased following miRNA depletion have a significantly higher density of miRISC clusters (10 and 15 min pulse, Figs 2E and EV2F, respectively).

In contrast to mRNAs, and in line with the observed changes in steady-state abundances, we found that the degradation rates of cytosolic lncRNAs are minimally impacted by miRNA depletion, with only a few displaying significant differences in degradation rate (10 and 15 min pulse, Figs 2D and EV2E). Specifically, most cytosolic lncRNAs behave similarly to nuclear lncRNAs (10 and 15 min pulse, Figs 2F and EV2G, respectively). The decrease in degradation rate between wild-type and miRNA-depleted cells is likely to be, at least in part a consequence of well-described compensation mechanisms (Dahan *et al*, 2011; Haimovich *et al*, 2013; Braun & Young, 2014) to account for decreased synthesis rates between the two cell types (Fig EV2H). The analysis of steady-state abundance and degradation rates following inducible loss of *Dicer* function indicate that, in contrast with coding transcripts, cytosolic lncRNAs are resilient to miRNA-mediated destabilization.

## Micropeptide-encoding transcripts undergo miRNA-dependent destabilization

Next, since our analysis of ribosomal profiling data indicated that a small fraction of cytosolic lncRNAs is ribosome-bound (Fig 1A), we investigated whether association with translating ribosomes would contribute to the impact of miRNAs on the degradation rates of some cytosolic lncRNAs. As expected, mRNAs are significantly more-efficiently translated than lncRNAs (Mann–Whitney *U*-test *P*-value = $2 \times 10^{-16}$). However, the translation efficiency of cytosolic lncRNAs, as a class, is significantly higher than that of nuclear lncRNAs (Mann–Whitney *U*-test *P*-value = $9 \times 10^{-5}$), indicating that some might encode micropeptides (Fig 3A). The short open reading frames of micropeptide-encoding transcripts are often missed by coding potential calculators, leading to the misclassification of these transcripts as lncRNAs (Makarewich & Olson, 2017). To distinguish *bona fide* lncRNAs from micropeptide-encoding transcripts, we used PhyloCSF (Lin *et al*, 2011) and identified 43 cytosolic transcripts

containing mammalian conserved short open reading frames (median longest predicted ORF length 216 nucleotides, Table EV1). These transcripts are almost three times more likely to be bound by ribosomes than are other cytosolic lncRNAs (Fig 3B), and their translation efficiency is significantly higher than that of cytosolic lncRNAs (two-tailed Mann–Whitney *U*-test *P*-value = $6 \times 10^{-5}$, Fig 3C), and more similar to that of mRNAs (two-tailed Mann–Whitney *U*-test *P*-value = 0.001, Fig 3C), which is consistent with the notion that some of these transcripts encode micropeptides. We separated micropeptides from *bona fide* cytosolic lncRNAs, and found that fold change in degradation rate of micropeptides in miRNA-depleted cells relative to control is similar to what is obtained for mRNAs, and significantly different from what is observed for *bona fide* lncRNAs (Mann–Whitney *U*-test *P*-value = 0.044, Fig 3D). These findings substantiate further potential the requirement of translation for miRNA-dependent transcript destabilization.

## miRNAs impact coding but not noncoding transcript stability

Our transcriptome-wide analysis indicates that translation is required for miRNA-dependent target destabilization. To test this hypothesis, we selected two cytosolic lncRNAs expressed at relatively different levels in mESCs: *TCONS_00034281* and *TCONS_00031378*, hereafter *lncRNA-c1* and *lncRNA-c2*, respectively (Fig EV3A–C). Both lncRNAs lack an apparent conserved ORF (Fig EV3A and B) and are weakly associated with ribosomes (Fig EV4A and B), supporting that they are noncoding. Consistent with our transcriptome-wide profiling experiment (paired two-tailed *t*-test *P*-value = 0.075 and 0.231, respectively, Fig EV3D), RT–qPCR analysis also supports that the steady-state abundance of endogenously expressed *lncRNA-c1* and *lncRNA-c2* is not significantly increased upon miRNA depletion (paired two-tailed *t*-test *P*-value = 0.564 and 0.221, respectively, Fig EV3E), in line with our hypothesis, and in contrast to *bona fide* miRNA targets such as *Lats2* or *Cdkn1A* (Wang *et al*, 2008; paired two-tailed *t*-test *P*-value = 0.028 and 0.040, respectively, Fig EV3D and E). Furthermore, and in contrast with *Lats2* or *Cdkn1A* (paired two-tailed *t*-test *P*-value = 0.044 and 0.032, respectively, Fig EV3F), stability of *lncRNA-c1* and *lncRNA-c2* is also not significantly affected in cells lacking DICER function (paired two-tailed *t*-test *P*-value = 0.132 and 0.240, respectively, Fig EV3F). Further substantiating our hypothesis, the steady-state abundance and stability for the two candidates is not affected by miRNA depletion, despite their cytosolic localization (Fig EV3G) and AGO2 binding, as supported by HEAP ((Li *et al*, 2020), Fig EV3A), AGO2-CLIP (Leung *et al*, 2011) data for mESCs, and as confirmed by AGO2-RIP (Fig EV3H and I) for *lncRNA-c1*, the most highly expressed of the two candidates. We attribute the lack of experimental evidence for *lncRNA-c2* association with AGO2 to this candidate's relatively low expression.

We reasoned that if translation is required for miRNA-dependent transcript destabilization, forcing association of the lncRNA candidates to translating ribosomes by fusing them downstream of a functional open reading frame should result in miRNA-dependent degradation of the fused transcripts (Fig 4A). We cloned *lncRNA-c1* and *lncRNA-c2* downstream of the Enhanced Green Fluorescent Protein stop codon (hereafter *GFP-lncRNA-c1* and *GFP-lncRNA-c2*, respectively), and transfected these constructs into wild-type and miRNA-depleted mESCs. As controls, we transfected in parallel

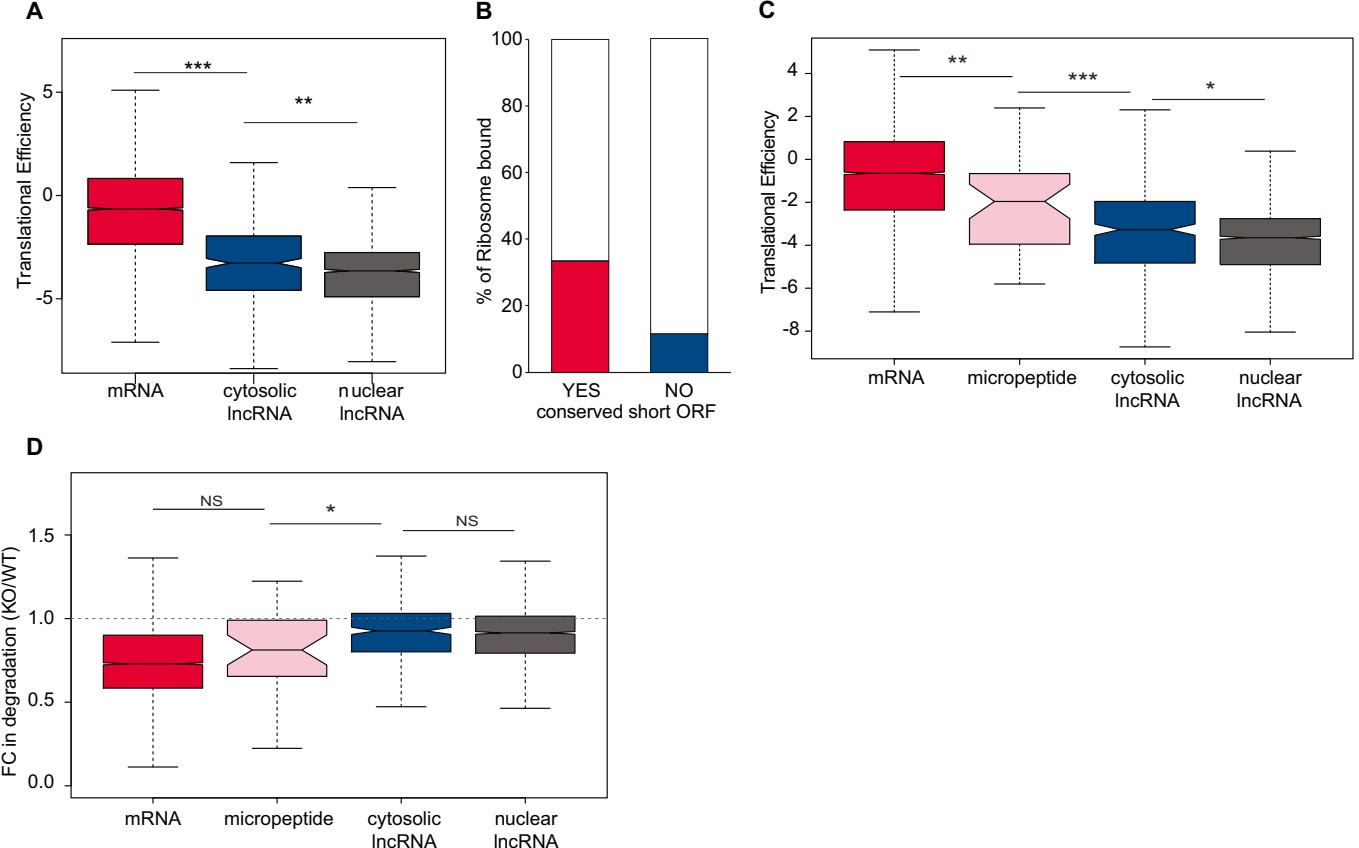

**Figure 3. Micropeptide-encoding transcript expression is post-transcriptionally regulated by miRNAs.**

A Distribution of the translational efficiency, in mESCs, of mRNAs ($n$ = 7,156, red), cytosolic ($n$ = 341, blue) and nuclear ($n$ = 1,915, grey) lncRNAs. Two-tailed Mann–Whitney $U$-test: **$P$-value < 0.01 and ***$P$-value < 0.001. Central band of boxplot represents median, box depicts 25–75 quantiles of distribution, and whiskers represent the 5th and 95th quantiles of the distribution.

B Fraction of cytosolic lncRNAs with experimental evidence for ribosomal binding with (red) or without (blue) an overlapping conserved short open reading frame.

C Distribution of the translational efficiency, in mESCs, of mRNAs ($n$ = 7,156, red), micropeptide-encoding transcripts ($n$ = 43, pink) and *bona fide* cytosolic ($n$ = 298, blue) and nuclear ($n$ = 1,857, grey) lncRNAs. Two-tailed Mann–Whitney $U$-test: *$P$-value < 0.05, **$P$-value < 0.01 and ***$P$-value < 0.001. Central band of boxplot represents median, box depicts 25–75 quantiles of distribution, and whiskers represent the 5th and 95th quantiles of the distribution.

D Distribution of the fold change (FC) in degradation rate of mRNAs ($n$ = 13,296, red), micropeptide-encoding transcripts ($n$ = 43, pink), *bona fide* cytosolic ($n$ = 759, blue) and nuclear ($n$ = 4,299, grey) lncRNAs in 4-OHT-treated (KO) relative to ethanol-treated (WT) cells after 8 days of treatment, horizontal dashed line represents a KO/WT FC in degradation rate of 1. Statistics: NS-$P$ > 0.05, *$P$-value < 0.05. Two-tailed Mann–Whitney $U$-test: $P$-value = 0.044. Central band of boxplot represents median, box depicts 25–75 quantiles of distribution, and whiskers represent the 5th and 95th quantiles of the distribution.

*GFP-*, *lncRNA-c1-* and *lncRNA-c2*-expressing constructs. The expression of exogenous *lncRNA-c1 and lncRNA-c2* decreased (Fig 4B) in miRNA-depleted cells and is therefore unlikely to be regulated by miRNAs. The levels of *GFP*, whose steady-state abundance is similar in wild-type and *Dicer*-depleted cells, also appear to not be post-transcriptionally regulated by miRNAs (Fig 4B). In contrast, and relative to their respective noncoding counterpart, *GFP-lncRNA-c1/c2* are significantly upregulated (paired two-tailed *t*-test, *P*-value = 0.012 and 0.034, respectively, Fig 4B) in miRNA-depleted cells, consistent with their miRNA-dependent destabilization in wild-type cells. Given that all constructs are under the control of the same promoter, the observed differences in expression are likely to be a direct consequence of changes in transcript stability.

Evidence of AGO2-binding to one of *lncRNA-c1's* miR-290/295 family of miRNAs response elements (MREs) supports that this candidate is bound by AGO2 loaded with one or more members

of this miRNA family (Fig EV5A and B and Appendix Fig S1). To validate that these miRNAs are indeed contributing to miRNA-dependent repression of *GFP-lncRNA-c1*, we co-transfected mESCs with *GFP-lncRNA-c1* expressing vector and miR-294-inhibitors. We note a significantly higher expression of *GFP-lncRNA-c1* in the inhibitor transfected cells compared to cells transfected with negative control (paired two-tailed *t*-test *P*-value < 0.006, Fig EV5C). We used site-directed mutagenesis to mutate three MREs for members of the miR-290/295 highly expressed miRNA family within *GFP-lncRNA-c1* (hereafter *GFP-lncRNA-c1ΔMRE*). As expected, reintroduction of miRNA mimics in DICER-depleted mESC significantly impacts the levels of the endogenous miRNA targets *Cdkn1a* (paired two-tailed *t*-test *P*-value = 0.024, Fig EV5D) and *Lats2* (paired two-tailed *t*-test *P*-value = 0.003, Fig EV5E). Consistent with the functionality of miR-290/295 MREs within *lncRNA-c1*, *GFP-lncRNA-c1* is significantly more downregulated upon miRNA mimic reintroduction than

*GFP-lncRNA-c1ΔMRE* (paired one-tailed *t*-test *P*-value = 0.043, Fig EV5F). Consistent with these MREs mediating, in part, *GFP-lncRNA-c1* miRNA-dependent repression in wild-type cells, we found that disrupting these miRNA-binding sites attenuated the

impact of miRNA depletion on GFP-lncRNA-c1 levels (paired two-tailed *t*-test *P*-value = 0.007, Fig 4C). We noted that *GFP-lncRNA-c1* is more responsive to global miRNA depletion than to MRE mutation. The presence of additional MREs for other mESC-expressed miRNAs

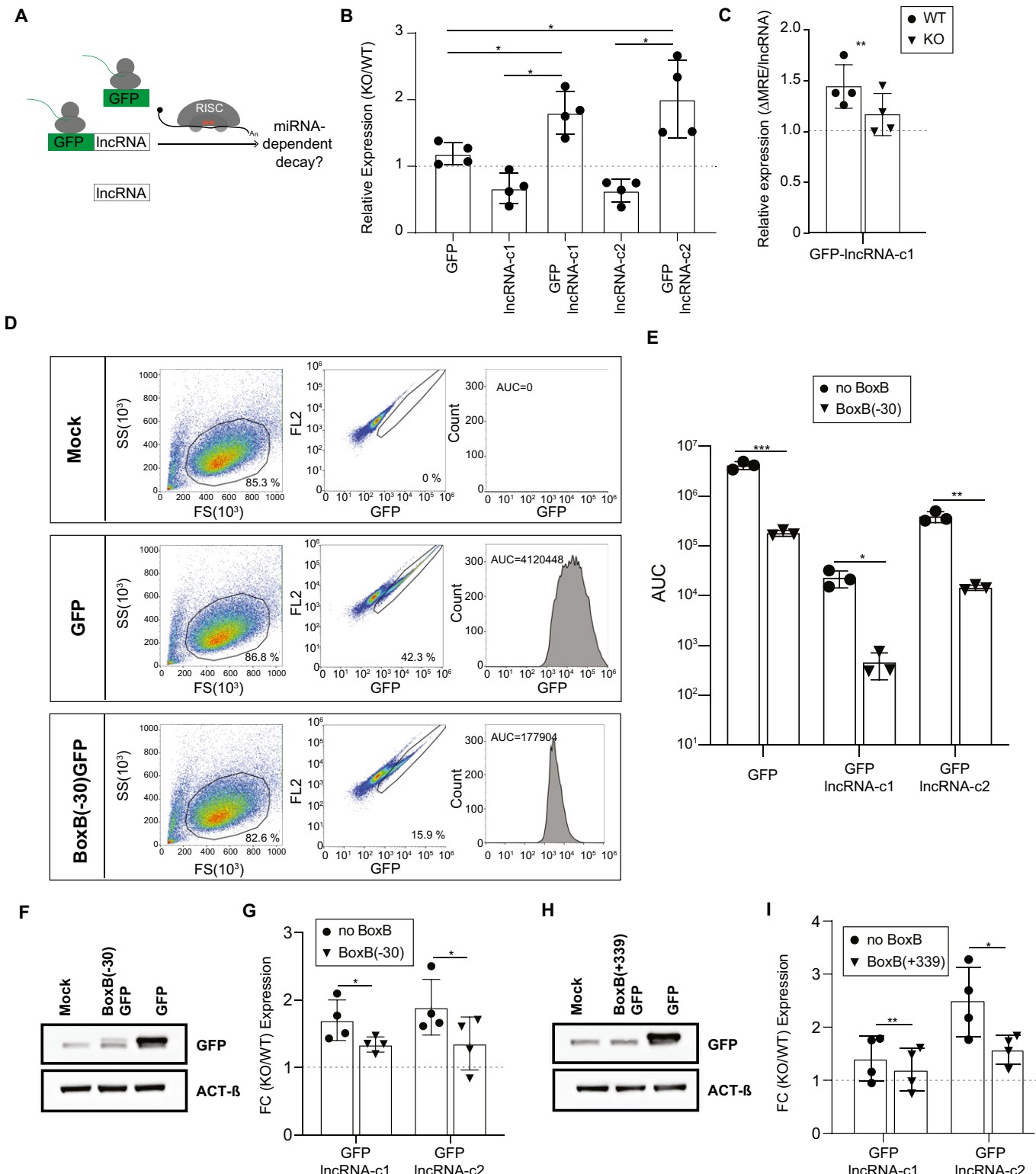

**Figure 4.**

Figure 4.   Association of lncRNA-c1 with translating ribosomes results in its miRNA-dependent decay.

A   Schematic of the construct tested in WT and miRNA-depleted mESCs.

B   Expression of *GFP*, *lncRNA-c1*, *GFP-lncRNA-c1*, *lncRNA-c2* and *GFP-lncRNA-c2* (x-axis) in 8 day 4-OHT-treated, miRNA-depleted cells (KO) relative to ethanol-treated (WT) mESC (y-axis) 24h post-transfection. Four independent biological replicates were treated, transfected and analysed by RT–qPCR. Statistical significance represented on the figure based on comparison of KO/WT fold change in expression of GFP, with *GFP-lncRNA-c1* and with *GFP-lncRNA-c2* (paired two-tailed *t*-test *P*-value = 0.034 and 0.032, respectively) and based on comparison of KO/WT fold change in expression of *GFP-lncRNAc1* with *lncRNA-c1* and *GFP-lncRNA-c2* with *lncRNA-c2* (paired two-tailed *t*-value = 0.012 and 0.018, respectively). Data are represented as mean ± SD, and each point corresponds to the results of one independent biological replicate.

C   Expression of *GFP-lncRNA-c1 ΔMRE* relative to *GFP-lncRNA-c1* (y-axis) in ethanol-treated (WT, circles) or 4-OHT-treated, miRNA-depleted cells (KO, triangles, x-axis). Four independent biological replicates were treated, transfected and analysed by RT–qPCR. Data are represented as mean ± SD, and each point corresponds to the results of one independent biological replicate. Paired two-tailed *t*-test *P*-value = 0.0071

D   GFP levels were determined using flow cytometry in mock, GFP and BoxB(−30)-GFP expressing cells 24 h post-transfection. For each row, in the left panel we represent side scatter intensity (SSC-A, y-axis) as a function of forward scatter intensity (FSC-A, x-axis). The percentage of gated events (cells) is show on the lower left corner. Centre panel represents GFP intensity (x-axis) as a function of FL2 (auto-fluorescence, y-axis). Percentage of GFP positive cells is shown on the lower right corner of the panel. Right panel represents the distribution of GFP fluorescence (x-axis) as a function of the number of cells (count, y-axis). The Area Under the Curve (AUC) is shown on the top left side of the panel.

E   AUC in mESCs expressing *GFP*, *GFP-lncRNA-c1* and *GFP-lncRNA-c2* wild-type construct (noBoxB, circles) or with a 5xBoxB cassette insertion 30 nucleotides upstream of the GFP start site (BoxB(−30), triangles). Comparison of AUC between construct with and without (BOxB(−30) paired two-tailed *t*-test *P*-value = $8.12 \times 10^{-4}$, 0.011 and 0.002 for *GFP*, *GFP-lncRNA-c1* and *GFP-lncRNA-c2*, respectively. Data are represented as mean ± SD, and each point corresponds to the results of one independent biological replicate.

F   Immunoblot analysis of GFP (GFP) in protein extracts from mESCs transfected with mock, BoxB(−30)-GFP and GFP expressing vectors. ACTIN-β (ACT-β) was used as an internal control. One representative blot is depicted.

G   Fold change (FC) in normalized expression of *GFP-lncRNA-c1*, *GFP-lncRNA-c2* (noBoxB, circles) and *BoxB(−30)-GFP-lncRNA-c1*, *BoxB(−30)-GFP-lncRNA-c2* (BoxB(−30), triangles; x-axis) 4-OHT-treated, miRNA-depleted mESCs (KO) relative to ethanol-treated mESCs (WT; y-axis). Four independent biological replicates were analysed. Data are represented as mean ± SD, and each point corresponds to the results of one independent biological replicate. Paired two-tailed *t*-test *P*-value = 0.0494 for GFP-lncRNA-c1 and *P*-value = 0.0355 for GPF-lncRNA-c2.

H   Immunoblot analysis of GFP (GFP) in protein extracts from mESCs transfected with mock, BoxB(+339)-GFP and GFP expressing vectors. ACTIN-β (ACT-β) was used as an internal control. One representative blot is depicted.

I   Fold change (FC) in normalized expression of *GFP-lncRNA-c1*, *GFP-lncRNA-c2* (noBoxB, circles) and *BoxB(+339)-GFP-lncRNA-c1*, *BoxB(+339)-GFP-lncRNA-c2* (BoxB(+339), triangles; x-axis) in 8 day 4-OHT-treated, miRNA-depleted mESCs (KO) relative to ethanol-treated mESCs (WT; y-axis). Four independent biological replicates analysed.

Data information: For all RT–qPCR analyses, transcript expression was first normalized by the amount of *Act-β* and *PolII* and next by the total amount of transfected vectors per cell estimated based on the levels of relative *Neomycin* expression. Each point corresponds to the results of one independent biological replicate. Statistics: NS- *P*-value > 0.05, **P*-value < 0.05, ***P*-value < 0.01 and ****P*-value < 0.001. Uncropped blots used for assembly of panels (F and H) are provided in Fig 4 Source Data. Source data are available online for this figure.

(14, Table EV2) likely explains why mutation of highly expressed miR290/295 seed-complementary MREs is not sufficient to entirely abolish miRNA-dependent *GFP-lncRNA-c1* destabilization.

If association with the translation machinery is sufficient to induce miRNA-dependent decay of a miRISC-bound noncoding transcript, one would expect translation inhibition of a protein-coding miRNA target to decrease its miRNA-induced decay. To test this, we inhibited *GFP* translation by inserting five BoxB sites (Eulalio *et al*, 2009) 30 nucleotides upstream of the GFP start codon (BoxB(−30)-GFP, Fig EV5G). Introduction of these hairpin structures in the GFP-5'UTR significantly decreased protein levels (Fig 4D–F). We refer to these constructs as BoxB(−30)-GFP-lncRNA-c1/c2. As previously described (Eulalio *et al*, 2009), we observed that 5xBoxB insertion in transcripts' 5'UTR is associated with decreased transcript steady-state levels (Fig EV5H). However, this up to 1.7-fold decrease in transcript level is not sufficient to explain the on average 67- and 27-fold reduction in GFP protein levels we observed for BoxB(−30)-GFP-lncRNA-c1/c2, respectively (Fig 4D and E and Appendix Fig S2).

We compared the impact of miRNA depletion on *GFP-lncRNA-c1/c2* with and without BoxB sites within their 5'UTR and found that translation inhibition significantly decreased the impact of miRNA depletion on *GFP-lncRNA-c1* and *GFP-lncRNA-c2* abundance (paired two-tailed *t*-test *P*-value = 0.049 and 0.0355, respectively, Fig 4G).

We next investigated whether the impact of translation on miRNA-dependent regulation was a consequence of inhibition of translation initiation or elongation. To distinguish between the two

processes, we generated a set of constructs, where we inserted 5xBoxB 339 nucleotides downstream of the GFP start codon (BoxB (+339)-GFP). To assess the impact of BoxB(+339) insertion on GFP translation, we used an antibody that targets the protein's N-terminus. This approach ensures that, even if insertion results in protein sequence changes downstream of the BoxB cassette, we should still be able to detect GFP. Similarly to what we observe for BoxB(−30) GFP (Fig 4D–G), translation is inhibited by BoxB(+339)-GFP (Fig 4H) and the decrease in transcript steady-state abundance associated with BoxB(+339) insertion (Fig EV5I) is not sufficient to explain the observed change in protein levels (Fig 4H).

Similarly to BoxB(−30), BoxB(+339)-GFP-lncRNA-c1/c2 levels are less impacted by miRNA depletion than are the levels GFP-lncRNA-c1/c2 (paired two-tailed *t*-test *P*-value = 0.003 and 0.030, respectively, Fig 4I). Given their position in the construct, we predict that BoxB(−30) and BoxB(+339) disrupt translation initiation and elongation, respectively. Since inhibition of translation elongation or initiation results in a similar reduction of miRNA-dependent regulation, we conclude that association with elongating ribosomes is required for miRNA-dependent transcript destabilization.

## Discussion

Post-transcriptional regulation by miRNAs leads to translational inhibition or transcript destabilization (Bartel, 2018). Whereas the

general consensus is that most miRNA-induced changes can be explained by transcript destabilization (Baek *et al*, 2008; Selbach *et al*, 2008), increasing evidence suggests that miRNA-dependent mRNA decay occurs co-translationally, raising questions about the ability of miRNAs to post-transcriptionally regulate the levels of noncoding transcripts.

Supporting different outcomes upon miRISC binding to coding and noncoding transcripts is recent evidence that these two classes of transcripts have distinct interaction dynamics with processing bodies (PB; Pitchiaya *et al*, 2019), the subcellular compartment where miRNA-dependent destabilization is thought to occur (Eulalio *et al*, 2007). Specifically, and in contrast to miRNA-bound mRNAs, which localize to the core of PB, miRNA-bound lncRNAs interact transiently and tend to locate to the PB periphery, a pattern that might reflect missing interactions with other molecular factors involved in miRNA-dependent regulation (Pitchiaya *et al*, 2019). One such factor could be DDX6, a PB-localized dead box helicase that links miRNA-dependent translation inhibition and decay (Chu & Rana, 2006; Chen *et al*, 2014; Rouya *et al*, 2014). In mESCs, loss of DDX6 function phenocopies loss of miRNA biogenesis (Baek *et al*, 2008; Freimer *et al*, 2018), suggesting that molecular factors that couple translation with RNA decay, like DDX6, are required for miRNA-dependent transcript destabilization.

These observations may appear surprising in light of previous reports that miRNA-dependent decay is minimally impacted by translation initiation or elongation (Wakiyama *et al*, 2007; Eulalio *et al*, 2009; Fabian *et al*, 2009), but one potential confounding factor in previous studies is their reliance on the use of exogenous reporters. Furthermore, the level of gene expression changes induced by the use of mimics or inhibitors exceeds the expected impact of miRNAs on their target levels by several-fold (Han *et al*, 2015; Seitz, 2019). Together these and other factors may limit the extent to which these earlier studies recapitulate what happens in vivo. Analysis of cytosolic *bona fide* lncRNAs, which have been previously shown to interact with miRISC (Helwak *et al*, 2013), but not with the translation machinery (Guttman *et al*, 2013), provides a unique opportunity to investigate the requirement of translation for endogenous miRNA-directed target decay. Furthermore, the relatively modest impact of miRNA depletion on target levels we observed following inducible DICER loss of function is in line with what would be expected for physiologically relevant miRNA-target interactions (Han *et al*, 2015; Seitz, 2019), thus supporting the use of this experimental system and the physiological significance of our findings.

Our transcriptome-wide analysis following miRNA loss revealed that, in contrast to mRNAs, steady-state abundance of cytosolic lncRNAs significantly decreased in miRNA-depleted cells, suggesting that this class of transcripts is not efficiently post-transcriptionally regulated by miRNAs. To assess the direct impact of miRNA regulation on cytosolic lncRNAs, we investigated, using RNA metabolic labelling, differences in the degradation rates of these transcripts in wild-type and miRNA-depleted cells. This analysis revealed that cytosolic lncRNA degradation rates decreased less than those of mRNAs, and to a similar extent as the degradation rates of nuclear lncRNAs, which are not expected to be regulated by miRNAs. The decrease of lncRNA degradation rates in miRNA-depleted cells is likely the result of coupling between RNA synthesis and decay, which has been proposed as a mechanism to ensure gene expression

homeostasis (Dahan *et al*, 2011; Haimovich *et al*, 2013; Braun & Young, 2014). While the decrease in degradation rates in response to decreased synthesis rates (Fig EV2H) is likely a general phenomenon in miRNA-depleted mESCs, the increased stabilization of coding transcripts in near-absence of miRNA is likely to obscure such effects for mRNAs.

Additionally, we show that the stabilities of putative micropeptides and mRNAs are similarly impacted by miRNAs, and more extensively than the stabilities of lncRNAs, further supporting the requirement of translation for miRNA-dependent regulation of endogenously expressed transcripts.

To validate this hypothesis, we selected two cytosolic lncRNAs, bound by AGO2 and with predicted binding sites for the miR-290/5 family, and forced their association to translating ribosomes by cloning them downstream of a functional protein-coding open reading frame. Consistent with the requirement of translation for miRNA-dependent transcript destabilization, forcing association to the ribosomes resulted in miRNA-dependent post-transcriptional regulation of previously unaffected transcripts (Fig 4B). These results are unlikely a consequence of pleotropic effects due to loss of miRNA function, as mutation of the putative MREs within candidate lncRNA sequence reduced the impact of miRNAs on candidate expression (Fig 4C). We conclude that miRNA-dependent regulation of endogenously expressed transcripts requires translation.

The requirement of translation for miRNA-dependent regulation supports that, despite extensive evidence for miRISC binding to cytosolic lncRNAs, the levels of these noncoding transcripts are not post-transcriptionally modulated by miRNAs. Evidence that miRNA-binding sites within lncRNAs evolved under constraint (Tan *et al*, 2015), suggests that miRNA-lncRNA interactions are biologically relevant. One possibility is that such interactions reflect miRNA-dependent regulation by lncRNAs. A number of examples support these roles in the context of disease and development (Cesana *et al*, 2011; Wang *et al*, 2013; Tan *et al*, 2014) and previous analysis of the potential extent of such regulatory roles by miRNAs suggested this mechanism of lncRNA function is prevalent among cytosolic transcripts (Tan *et al*, 2015). However, given the relatively low abundance of most lncRNAs, which rarely exceeds the expected threshold to exert significant and physiological relevant changes in miRNA targets (Bosson *et al*, 2014; Denzler *et al*, 2014; Hausser & Zavolan, 2014), the biological relevance of miRNA-dependent regulation by lncRNAs remains controversial. In light of the present results that support a different outcome of miRNA interactions with mRNA or lncRNAs, further experiments are now needed to assess the generality of mRNA-based conclusions.

More generally, the present results also imply that miRISC binding per se is not sufficient to determine the outcome of bound targets, suggesting the requirement of further, yet unidentified, molecular partners. For example, GW182, a major component of miRISC, recruits the CCR4-NOT deadenylase complex (Braun *et al*, 2011; Fabian *et al*, 2011) and deadenylation of mRNAs by CCR4-NOT has been shown to be a key step in facilitating miRNA-mediated transcript decay (Tucker *et al*, 2001; Denis & Chen, 2003). Understanding the differences in coding and noncoding RNA miRNA-dependent deadenylation, for example by genome-wide investigation of poly(A)-tail length following DICER depletion, or identifying translation-dependent factors, may provide

the much-needed mechanistic insights on how translation of targets facilitates miRNA-dependent gene regulation.

In summary, the analysis of endogenously expressed and miRISC-bound noncoding transcripts provides further evidence that translation is indispensable for miRNA-dependent regulation of endogenous transcripts, suggesting the requirement of further molecular partners, and highlighting differences in post-transcriptional regulation of coding and noncoding RNAs.

# Materials and Methods

Detailed description of Resources and Tools used in this study can be found in Appendix Table S1.

## Mouse embryonic stem cell culture

Feeder depleted mouse DTCM23/49 XY embryonic stem cells (Nesterova *et al*, 2008; Tan *et al*, 2015; Graham *et al*, 2016) were grown on 0.1% gelatin-coated tissue culture-treated plates in a humidified incubator with 5% (v/v) $CO_2$ at 37°C in 1 × DMEM medium supplemented with 1 × nonessential amino acids, 50 μM β-mercaptoethanol, 15% foetal bovine serum, 1% penicillin/streptomycin and 0.01% of recombinant mouse leukaemia inhibitory factor. Cultures were maintained by passaging cells every 48 h (replating density ~3.8 × $10^4$ cells/cm$^2$). Unless stated otherwise, to induce loss of *Dicer* function, cells were cultured in mESC growth media supplemented with 800 nM tamoxifen and previously resuspended in 100% ethanol ([Z]-4-Hydroxytamoxifen [4-OHT]) for 48 h. To account for indirect effects of ethanol addition to cells, equal concentration of 100% ethanol was added to control non-miRNA-depleted DTCM23/49 XY embryonic stem cells for 48 h. Subsequently, miRNA-depleted and non-miRNA-depleted cells were transferred to nonsupplemented mESC growth medium and cultured for six additional days to deplete miRNA levels in 4-OHT-treated DTCM23/49 XY embryonic stem cells.

## Small RNA extraction in Dicer depletion time course

Feeder depleted mouse DTCM23/49 XY embryonic stem cells were cultured in mESC growth media supplemented with 800 nM tamoxifen 4-OHT. Small RNA extraction and DNAse treatment following 0, 4, 8, 10 and 12 days of 4-OHT treatment were performed using the Qiagen miRNEasy Mini Kit and Qiagen RNAse-free DNAse according to the manufacturer's instructions.

## Small RNA sequencing, mapping and quantification

Small RNA libraries were prepared from 500 ng of total RNA using Illumina TruSeq small RNA protocol and sequenced on Illumina HiSeq 2500.

Sequencing adapters were removed from fifty nucleotides long single-end reads using cutadapt (v1.8) and mapped to mouse genome (mm10) using bowtie2 (v2.2.4). Gene expression levels for all mouse miRNAs annotated in miRbase (v21; Kozomara *et al*, 2019) were quantified using HT-seq (v0.6.1). The raw sequencing data and reads counts are available on the NCBI Gene Expression Omnibus (GEO) under accession number GSE143277.

## Western blot analysis

Approximately 500,000 mESCs were harvested and washed twice with ice-cold PBS and stored, after PBS removal, at −80°C until lysis. Lysis was performed by incubating previously frozen cell pellet in 50 μl of cold RIPA Buffer (150 mM NaCl, 1.0% NP-40, 0.5% sodium deoxycholate, 0.1% SDS, 50 mM Tris, pH 8.0) on a rotating wheel for 1 h at 4°C. Lysed cells were sonicated using a Bandelin Sonopuls HD 2070 homogenizer to degrade released DNA (Power = 40%, 20 s ON/OFF cycles for three total sonication pulses). Protein concentration was determined using the Pierce™ BCA Protein Assay kit according to the manufacturer's instructions.

For NANOG, OCT-4 and DICER protein level quantification, 30 μg of protein were separated at 110 V on NuPage™ 12% Bis-Tris gel and transferred overnight at 4°C and 27 V in transfer buffer (25 mM Tris–HCl pH 7.6 192 mM glycine, 20% methanol) onto nitrocellulose membranes. Transfer efficiency was assessed by staining the membrane with Ponceau S solution and staining solution was subsequently removed by washing the membrane three times with TBS-T (Tris-buffered saline, 0.1% Tween 20, 5 min, room temperature) After incubation with 5% skim milk in TBS-T for 4–6 h at 4°C, the membranes for simultaneous NANOG and DCR detection, membranes were cut at 100 kDa (Fig EV1 Source Data) washed once in TBS-T and incubated with anti-DICER (1:4,000), anti-NANOG (1:1,000) antibodies in 5% skim milk in TBS-T overnight at 4°C on a see-saw shaker. For OCT-4 detection membrane was incubated with anti-OCT4 (dilution 1:1,000) antibodies in 5% skim milk in TBS-T overnight at 4°C on a see-saw shaker. Following primary antibody incubation, membranes were washed three times for 20 min in TBS-T and subsequently incubated with Secondary Antibodies (DICER = 1:4,000 Goat Anti-Rabbit IgG (H + L)-HRP Conjugate; for NANOG = 1:4,000 Goat Anti-Rabbit IgG (H + L)-HRP Conjugate; for OCT-4 = 1:2,500 Rabbit Anti-Goat IgG/HRP)). Next, membranes were washed three times for 20 min in TBS-T and chemiluminescent detection was performed using the Advansta ECL Western Bright as per the manufacturer's instructions.

Following detection, secondary antibody coupled with the HRP was deactivated by washing the membrane two times for 20 min with 1% (w/v) Sodium-Azide in TBS-T and the membrane incubated 2 h at 4 °C with 5% (w/v) skimmed milk in TBS-T containing anti-ACTIN-β loading control diluted 1:4,000. The membranes were subsequently washed three times for 15 min in fresh TBS-T and incubated for one hour at room temperature with the secondary antibody coupled with horseradish peroxidase in 5% skimmed milk in TBS-T (for ACTIN-β = 1:4,000 Goat Anti-mouse IgG). Following washing of membranes three times for 15 min in fresh TBS-T, ACTIN-β protein detection was performed using the Advansta ECL Western Bright as per the manufacturer's instructions. For uncropped images of blots, please refer to Fig EV1 Source Data.

For GFP Western blot analysis, we quantified protein levels in cells harvested 72 h following construct transfection. Following lysis, sonication and BCA quantification (performed as described for NANOG, OCT-4 and DICER protein level quantification), 15 μg of protein was separated at 110 V on NuPage™ 12% Bis-Tris gel and transferred for one hour at 4°C at 100 V in transfer buffer (25 mM Tris–HCl pH 7.6 192 mM glycine, 20% methanol) onto nitrocellulose membranes. Transfer efficiency was assessed by staining with Ponceau S solution as described and staining solution was removed

by washing the membrane three times with TBS-T (Tris-buffered saline, 0.1% Tween 20, 5 min, room temperature). Subsequently, the membrane was horizontally cut at 35 kDa to allow for simultaneous probing of GFP (27 kDa) and ACTIN-β loading control (42 kDa). Blocking was performed for 1–2 h in 5% (w/v) skimmed milk in TBS-T and primary antibody incubation was performed overnight at 4°C (rabbit anti-GFP = 1:1,000 and mouse anti-ACTIN-β = 1:4,000 antibody dilution in 5% (w/v) skimmed milk in TBS-T). Subsequently, membranes were washed three times for 20 min in TBS-T and incubated with Secondary Antibodies (GFP = 1:1,000 Goat Anti-Rabbit IgG (H + L)-HRP Conjugate; ACTIN-β = 1:4,000 Goat Anti-mouse IgG). Following secondary antibody incubation membranes were washed three times for 15 min with fresh TBS-T and detection was performed as described above. For uncropped images of blots, please refer to Fig 4 Source Data.

### Cell proliferation assay

16–24 h prior to DNA staining, 33,000 cells/cm$^2$ were plated on a 6-well gelatin-coated tissue culture plate. Edu (Click-iT Edu Alexa Fluor™ 488 Flow Cytometry Assay Kit) was added to mESC growth medium at a final concentration of 10 μM, and the cells were incubated at 37°C for 30 min in a humidified incubator with 5% (v/v) $CO_2$. Cells were trypsinized, counted and, for each tested sample, 750,000 cells were washed once with 3 ml of 1% BSA in PBS, resuspended in 100 μl of Click-iT fixative buffer and incubated for 15 min at room temperature in the dark. Cells were washed with 3 ml of 1% BSA in PBS, centrifuged and the supernatant removed. The pellet was resuspended in 100 μl of 1 × Click-iT saponin-based permeabilization and wash reagent, and the cells incubated for 15 min at room temperature in the dark. 500 μl of freshly prepared Click-iT reaction cocktail containing Alexa Fluor 488 Fluorescent dye Azide was added to the permeabilized cells in 1 × Click-iT saponin-based permeabilization and wash reagent and the mix incubated at room temperature in the dark for 30 min. Cells were washed once with 3 ml of 1 × Click-iT saponin-based permeabilization and wash reagent and following supernatant removal resuspended in 500 μl of Click-iT saponin-based permeabilization and wash reagent. Cells were analysed by flow cytometry on a Beckman Coulter Gallios Flow Cytometer according to the manufacturer's instructions, using a 488 nm excitation wavelength and a green emission filter (530/30 nm).

### 4sU metabolic labelling

Five million DTCM23/49 XY mESCs (WT and miRNA-depleted) were seeded and allowed to grow to 70–80% confluency (approximately 1 day). 4sU was added to the growth medium (final concentration of 200 μM) and cells were incubated at 37 °C for 10 or 15 min. RNA was extracted using TRIzol according to the manufacturer's instructions and DNAse-treated using RNeasy on-column digestion according to the manufacturer's instructions. Hundred microgram of RNA was incubated for 2 h at room temperature with rotation in 1/10 volume of 10 × biotinylation buffer (Tris–HCl pH 7.4, 10 mM EDTA) and 2/10 volume of biotin-HPDP (1 mg/ml in dimethylformamide). Following biotinylation, total RNA was purified through phenol:chloroform:isoamyl alcohol extraction and precipitated with equal volume of Isopropanol and 1/10 volume of 5 M NaCl. RNA was washed once with 75% ethanol and

resuspended in DEPC-treated H$_2$O. Equal volume of biotinylated RNA and pre-washed Dynabeads™ MyOne™ Streptavidin T1 beads were mixed and incubated at room temperature for 15 min under rotation. The beads were then separated using a DynaMag™-2 Magnetic stand. The supernatant (that contains unlabelled pre-existing RNA) was placed at 4°C until precipitation. Beads were washed 3 × with 1 × B&W Buffer (5 mM Tris–HCl pH 7.5, 0.05 mM EDTA, 1 M NaCl in DPEC-H$_2$O) and biotinylated RNA dissociated from streptavidin-coated beads by treatment with 100 mM 1,4-Dithiothreitol for 1 min, followed by 5 min in RTL buffer. Beads were separated from the solution using DynaMag™-2 Magnetic stand and the RNA recovered from the supernatant extracted using Qiagen RNeasy Mini Kit according to the manufacturer's instructions. Pre-existing RNA was precipitated with equal volume of Isopropanol and centrifuged for 45 min at 15,000 $g$ at 4°C. Pre-existing RNA pellet was washed with 75% ethanol and resuspended in DEPC-treated H$_2$O. Metabolic labelling experiments were repeated once for the two labelling durations (2 independent 4-OHT- and ethanol-treated biological replicates).

### RNA sequencing, mapping and quantification of metabolic rates

Total RNA libraries were prepared from 10 ng of DNase-treated pre-existing and newly transcribed RNA using Ovation® RNA-Seq and sequenced on an Illumina HiSeq 2500 (average of fifty million reads per library).

Hundred nucleotides long single-end reads were first mapped to *Mus musculus* ribosomal RNA (rRNA, ENSEMBL v91, Cunningham *et al*, 2019) with STAR v2.5.0 (Dobin *et al*, 2013). Reads that do not map to ribosomal RNA were then aligned to intronic and exonic sequences of *Mus musculus* transcripts database (ENSEMBL v91) using STAR and quantified using RSEM (Li & Dewey, 2011). Principal Component Analysis (PCA) of read counts was performed to demonstrate separation between newly transcribed (labelled) and total RNA (Fig EV2A–C). Rates were inferred, independently at each labelling point using the INSPEcT ([35] Bioconductor package v1.8.0). Specifically, the absolute values of synthesis, processing and degradation rates in each condition were estimated using the "newINSPEcT" function with the option pre-existing = TRUE, while the statistical significance of the variation of the rates between conditions was obtained using the method "compareSteady" [see INSPEcT vignette at http://bioconductor.org/packages/INSPEcT/]. The raw sequencing data are available on the NCBI Gene Expression Omnibus (GEO) under accession number GSE143277 (Li & Dewey, 2011; Dobin *et al*, 2013).

### Identification of AGO2 bound regions in mESCs

AGO2 bound regions in mESCs were downloaded from (Leung *et al*, 2011).

Cutadapt (Martin, 2011) was used to remove sequence adapters from publicly available AGO2-CLIP sequencing reads from wild-type and *Dicer* mutant mESCs (Leung *et al*, 2011). Trimmed reads were mapped to the mouse genome (mm10) using bowtie (Langmead *et al*, 2009; bowtie -v 2 -m 10 --best –strata) as previously described (Corcoran *et al*, 2011). Mapped reads from the same cell type were merged AGO2 bound clusters identified using PARAlyzer v1.5 (Bandwidth = 3; minimum read count per group = 5; minimum read count per cluster = 1; minimum read count for KDE = 5;

minimum cluster size = 1; minimum conversion count per cluster = 1; minimum read count for cluster inclusion = 1; Corcoran *et al*, 2011). Clusters present in wild-type and DICER null cells were excluded using BEDtools (Quinlan & Hall, 2010).

### Translational efficiency

Ribosome profiling (RP) and total RNA raw reads were downloaded from SRA database (SRX084815 and SRX084812, respectively (Ingolia *et al*, 2011)). Reads were trimmed based on quality and sequence adapters removed with Cutadapt (v. 1.8, Martin, 2011). Only reads with the expected read length (16 to 35 nt for the ribosome footprint and 35 to 60 nt for total RNA) were kept for further analysis. Reads mapping *Mus musculus* ribosomal RNA (rRNA) and transfer RNA (tRNA) databases (ENSEMBL v91, Cunningham *et al*, 2019) using to bowtie2 (v. 2.3.4.1, parameters: -L 15 -k 20, Langmead & Salzberg, 2012) were excluded. The remaining reads (SRX084815: 12 228 002 reads; SRX084812: 12 361 681 reads) were aligned against *Mus musculus* transcripts database (ENSEMBL v91) using bowtie2 (v. 2.3.4.1, -L 15 -k 20). Multi-mapping reads (mapping to 2 or more transcripts from different gene loci) were filtered out and the remaining reads summarized at a gene level using an in-house script. Translational efficiency (TE) was calculated in R. Briefly, raw genes ribosome footprints and total RNA counts were normalized using the edgeR package to account for variable library depths (cpm function; Robinson *et al*, 2010). Translational efficiency (TE) was calculated as the log2 ratio between normalized RP counts and normalized TR counts. TE was only calculated for genes with cpm > 1 and have RP read > 0.

Conserved short open reading frames within lncRNA transcripts were identified by overlapping lncRNA loci with regions with positive PhyloCSF scores, those that likely represent conserved coding regions, in any of the three possible reading frames on the same strand as the lncRNA transcript (Lin *et al*, 2011). LncRNA transcripts containing conserved short open reading frames are likely to encode micropeptides.

### Subcellular fractionation

Subcellular fractionation of mESCs was carried out using the PARIS kit according to the manufacturer's instructions. Following RNA extraction from cytosolic and nuclear fractions, genomic DNA was removed from samples using TURBO DNAse according to the manufacturer's instructions. DNAse-treated RNA was extracted using phenol chloroform and RNA precipitated using equal volume of isopropanol and 1/10 volume of 5 M NaCl. RNA pellet was washed once with cold (4°C) 75% ethanol in DEPC-H$_2$O and resuspended in DEPC-treated H$_2$O.

### RNA extraction and qPCR

Total cellular RNA was extracted with the Qiagen RNeasy Mini kit according to the manufacturer's instructions. To quantify levels of mature miRNAs, total RNA was extracted with the Qiagen miRNeasy kit. Genomic DNA was removed by performing a column Qiagen DNAse I treatment according to the manufacturer's instructions. Following RNA elution in DEPC-H$_2$O, an additional DNAse treatment was performed using TURBO™ DNAse according to the

manufacturer's instructions and RNA was purified and reprecipitated as described in preceding section. Following precipitation, RNA was reverse-transcribed using the Qiagen Quantitect Reverse Transcription Kit. Quantitative PCRs were prepared using the Roche FastStart DNA Essential DNA Green Master and sequence-specific primers (Appendix Table S2) and analysed using a Roche Light Cycler®96. Unless otherwise stated *Actin-β* and *PolymeraseII* were used as internal controls.

For miRNA level quantification, RNA was reverse-transcribed using the Applied Biosystems TaqMan microRNA Reverse Transcription Kit and small RNA-specific probes for miR-294-3p, miR-290-3p and sno-202 according to the manufacturer's instructions. Small RNA expression levels relative to *small nucleolar RNA 202* (sno-202) were subsequently quantified on a Roche Light Cycler®96 using the TaqMan Universal Master Mix II, no UNG, according to the manufacturer's instructions.

### RNA stability

Transcription was inhibited by adding actinomycin-D and resuspended in dimethyl sulphoxide at a final concentration of 10 μg/ml in supplemented mESC growth medium. Stability of transcripts was inferred by comparing relative gene expression levels (normalized to *Actin-β*) in cells incubated for 8 h with actinomycin-D and untreated cells.

### Candidate lncRNA and mRNA analysis

Enhanced Green Fluorescent Protein gene was amplified from the pBS-U6-CMV-EGFP plasmid (Sarker *et al*, 2005) with primers complementary to EGFP and NheI restriction sites (see Appendix Table S2) and inserted into NheI digested pcDNA3.1(-) (Addgene, V79520). Ligation was performed using T4 DNA ligase according to the manufacturer's instructions. Plasmid was transformed into DH5α subcloning efficiency bacterial cells and Sanger sequencing was used to confirm correct orientation of EGFP insertion into plasmid (*GFP*).

*lncRNA-c1* and *lncRNA-c2* were amplified from mESC cDNA using sequence-specific primers with overhangs containing restriction sites for either XhoI or EcoRI (Appendix Table S2) and cloned directionally into XhoI-EcoRI digested pcDNA3.1(-) plasmid to generate *lncRNA-c1* construct downstream of the CMV and T7 promoter. Ligation was performed using T4 DNA ligase according to the manufacturer's instructions. *GFP-lncRNA-c1* construct was generated adopting same cloning strategy but inserting *lncRNA-c1* into *GFP* containing pcDNA3.1(-) construct. Sanger sequencing was used to confirm correct sequence.

One MRE on *GFP-lncRNA-c1* was mutated using the Phusion High-Fidelity Polymerase. Briefly, primers containing a scrambled sequence of the seed region within the MRE and wings complementary to the targeted sequence (Appendix Table S2) were used to amplify from the *GFP-lncRNA-c1* containing plasmid. Following amplification PCR purification was performed and DpnI digestion was used to digest template plasmid. Blunt end ligation using T4 DNA ligase was performed to ligate amplified sequence containing mutated MRE according to the manufacturer's instructions. Ligated construct was subsequently transformed into DH5α bacterial cells and MRE mutation was confirmed through Sanger sequencing.

The remaining two MREs on *GFP-lncRNA-c1* were done using the Takara In-fusion HD cloning kit according to the manufacturer's instructions. The primers were designed using the manufacturer online design tool (https://www.takarabio.com/learning-centers/cloning/in-fusion-cloning-tools) and are available in Appendix Table S2. The MREs were mutated sequentially using primers containing the mutation of interest and AmpHiFi PCR Master Mix. PCR products were gel purified, ligated using the In-fusion HD enzyme and transformed into Stellar competent bacterial cells according to the manufacturer's instruction. MRE mutation was confirmed through Sanger sequencing.

Five BoxB hairpins were amplified from pAc5.1C-5BoxB73-Fluc-STOP-CG10011-SV40 plasmid (gift from Dr. Heike Budde at the Max Planck Institute of Tuebingen, Germany) and inserted 30 nucleotides upstream and 339 nucleotides downstream of the start codon (BoxB(−30) and BoxB(+339), respectively) in *GFP*, *GFP-lncRNA-c1* and *GFP-lncRNA-c2* using Takara in-Fusion HD cloning kit according to the manufacturer's instructions. Primers were designed using the manufacturers online tool and are available in Appendix Table S2. PCR products were gel purified using NucleoSpin gel and PCR clean-up kit (Macherey-Nagel), ligated using In-fusion HD enzyme and transformed into DH5α bacterial cells. Proper insertion of the hairpins was confirmed through Sanger sequencing.

For all transfected lncRNA candidate constructs, analyses in wild-type and miRNA-depleted mESCs, 6 days of ethanol (WT) and 4-OHT-treated (miRNA-depleted, KO) DTCM23/49 XY embryonic stem cells were plated in 10 cm dishes at a density of 35,000 cells/$cm^2$. The following day, cells were transfected with $484 \times 10^{-15}$ mol of candidate expressing vector using the lipofectamine 2000 transfection reagent. RNA was extracted 24 h after transfection (day 8 of ethanol and 4-OHT treatment for WT and KO mESC, respectively). Gene expression levels relative to *Actin-β* and *PolymeraseII* of transfected candidates were normalized to *Neomycin* expression to account for differences in transfection efficiency between different cell types and experiments.

For miRNA mimic and inhibitor transfections mmu-miR294-3p, mmu-miR295-3p mimics (100 nM) and mmu-miRNA294-3p inhibitors (5, 15 and 30 mM), and miRNA mimic/inhibitor negative controls were transfected 24 h after plasmid transfection using the RNAimax transfection reagent according to the manufacturer's instructions. RNA was extracted 24 h after small RNA transfection (day 9 of ethanol and 4-OHT treatment for WT and KO mESC, respectively), and reverse transcription was performed according to the manufacturer's instructions as described above.

**Flow cytometry analysis**

One day before transfection, 250,000 wild-type DTCM23/49 XY embryonic stem cells were plated in 6 well plates. Cells were transfected using 9ul of lipofectamine 2000 and 48.5 fmol of the following vectors per well: pcDNA3.1(-) as mock, *GFP*, *BoxB(−30)-GFP*, *GFP-lncRNA-c1*, *BoxB(−30)-GFP-lncRNA-c1*, *GFP-lncRNA-c2* and *BoxB(−30)-GFP-lncRNA-c2*. Twenty-four hours post-transfection, cells were trypsinized, washed once with PBS and being resuspended in PBS with 10% FBS and 2 mM EDTA. Cells were analysed on Gallios flow cytometer (Beckman Coulter) and physical parameters (forward scatter, FSC and side scatter, SSC) as well as information from FL1 (GFP, 525 nm emission) and FL2 channels

(auto-fluorescence, 575 nm emission) was collected. Analysis of flow cytometry data was done using FlowJo (v10.6.1). Primary gating using FSC and SSC was set to exclude dead cells and debris, FL1 and FL2 channels were used to gate on GFP$^+$ cells. Number of cells (Cell count) at each FL1 (GFP) intensity was exported to GraphPad Prism (v8.4.3) and used to calculate the area under the curve (AUC).

**RNA immunoprecipitation**

RNA immunoprecipitation was performed as previously described [66]. Briefly, $4.8 \times 10^6$ E14 WT cells were seeded into 10 cm dishes 16 h prior to harvest. At the same time, 60 μl of Protein A/G Plus-Agarose beads were incubated with 10 μl of Rabbit anti-AGO2 or 2.5 μg of normal rabbit IgGs. Protein content in cell lysate was split in half, adjusted to 1 ml using IP Lysis buffer and supplemented with protease inhibitors and RNase inhibitors. Fifty microliter of diluted cell lysates were collected for Input (5%). The remaining cell lysate was added to the A/B or IgG coupled beads and incubated overnight at 4°C, on a rotating wheel. After washing, 100 μl of RIP buffer + 1 μl of RNAse inhibitor was added to the beads and centrifuged. Twenty microliter and 80 μl of supernatant were collected for protein and RNA analysis. Immunoprecipitated and input RNA was extracted using TRIzol reagent, resuspended in DNAse reaction mix (16 μl ddH$_2$O, 2 μl 10 × RQ1 DNase buffer, 2 μl RQ1 DNase) and reverse-transcribed using the GoScript RT Kit and oligo d[T]$_{18}$. RT–qPCR were performed as described above.

Ten microliter of RIP supernatant and input samples were separated on 8% SDS–PAGE gels and transferred to PVDF membranes. After incubation with 5% skim milk in 1 × PBS/0.1% Tween-20, the membranes were washed and incubated with antibodies against AGO2 (ARGONAUTE 2 Rabbit mAb) and Dicer (Rabbit anti-Dicer) at 4°C for 16 h. Secondary antibodies (anti-Rabbit IgG-HRP) were incubated on membranes for 1 h at RT at a dilution of 1:5,000. Immunoblots were developed using the SuperSignal West kit and detected using an imaging system. Membrane stripping was performed by low pH method and AGO2 membrane was re-probed with antibodies against AGO2 (Argonaute 2 Mouse mAb). All membranes were stained using a coomassie blue staining solution to ensure equal loading.

# Data availability

The datasets produced in this study are available in the following database

- Total, 4sU labelled and small RNA-Seq datasets: Gene Expression Omnibus GSE143277 (https://www.ncbi.nlm.nih.gov/geo/query/acc.cgi?acc=GSE143277)

**Expanded View** for this article is available online.

# Acknowledgements

We would like to thank the Genomics Technology Facility at the University of Lausanne for help with RNA integrity analysis, library preparation and sequencing.

We would like to thank Maria Ferreira Da Silva for performing the flow cytometry analysis of the proliferation rate assay and Alaadin Bulak Arpat for helpful discussions on the ribosomal profiling analysis. We would like to thank

Dr. Heike Budde from the Max Planck Institute of Tuebingen, Germany, for supplying us with pAc5.1C-5BoxB73-Fluc-STOP-CG10011-SV40 plasmid. The computations were performed at the Center for High-performance Computing of the University of Lausanne. This work was funded by the Swiss National Science Foundation (grant PP00P3_150667 to A.C.M and 31003A_173120 to C.C) and the NCCR in RNA & Disease (A.C.M. and C.C).

## Author contributions

AB and ACM designed the study. AB, SdP, JYT, RD and ACM performed the *in silico* analysis. AB, BA, RA, HW, CC and ACM performed and analysed *in vitro* experiments analysis. MP, CC and ACM supervised the study. ACM wrote the manuscript. All coauthors read and approved the manuscript.

## Conflict of interest

The authors declare that they have no conflict of interest.

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
