## [Review Process File · The EMBO Journal]

Translation is required for miRNA-dependent decay of endogenous transcripts.

Adriano Biasini, Baroj Abdulkarim, Stefano De Pretis, Jennifer Tan, Rajika Arora, Harry Wischniewski, Rene Dreos, Mattia Pelizzola, Constance Ciaudo, and Ana Claudia Marques

DOI: N/A

Corresponding author(s): Ana Claudia Marques (anaclaudia.marques@unil.ch)

Review Timeline:

Submission Date:	28th Jan 20
Editorial Decision:	5th Mar 20
Revision Received:	3rd Sep 20
Editorial Decision:	12th Oct 20
Revision Received:	30th Oct 20
Accepted:	6th Nov 20

Editor: Stefanie Boehm

Transaction Report:

Dear Dr. Marques,

Thank you for submitting your manuscript proposing a requirement for translation in miRNA-mediated mRNA decay for consideration by The EMBO Journal. We have now received three reports on your study, which are included below for your information.

As you will see, the referees appreciate that your study addresses an important question in the field. However they are also not yet convinced that the conclusions and the proposed model are sufficiently supported by the current set of experiments and find that more mechanistic understanding would be needed. In particular, experiments that demonstrate that ribosome binding and/or translation are required for miRNA-mediated mRNA decay would be crucial (ref#1- point 3; ref#2- point 4; ref#3- point 4), as well as further validation and additional control experiments (ref#-1 point 1, 2; ref#3- point 1, 2, 3). Should you be able to fully address these key concerns, then we would be happy to consider the manuscript further for publication at EMBO Journal. Please however note that our policy allows only a single round of major revision. We recognize that addressing the referees' comments fully and in the depth that would warrant further consideration here, would require a significant amount of experimental work and time. In addition, this will likely include experiments with an unforeseeable outcome and thus potentially go beyond the extent of a normal revision. Even though we can extend revision time up to a total of six months in certain cases, it is important to clarify all concerns at this stage before committing to a revision. Alternatively, we could also offer to discuss a potential transfer of the manuscript with referee reports to EMBO reports, which will have different requirements regarding the extent of mechanistic insight that needs to be provided.

Referee #1:

Biasini et al.

Translation is required for miRNA-dependent decay of endogenous transcripts

In this study, the authors have analyzed genome-wide expression profiling data to correlate transcript levels with translation-associated, miRNA-mediated mRNA decay. Based on available data sets, the authors separated cytosolic and nuclear fractions on lncRNAs. Ribosome profiling as well as Ago2-CLIP data, which are also publicly available revealed that lncRNA rarely associate with ribosomes but do interact with miRISC. Inactivation of miRNA expression by Dicer-deletion in mESCs revealed that steady-state levels of lncRNAs are only marginally affected by miRNA expression, while mRNA levels increased as expected when Dicer is deleted. Using additional statistical tests, the authors report that miRISC binding to lncRNAs has not much impact on the expression levels. However, when they analyzed the small population of lncRNAs that contain short open reading frames and have the potential to generate micropeptides, they find that those lncRNAs are affected by the loss of miRNA expression. The authors validate their model by fusing a lncRNA to a GFP reporter and demonstrate that miRNA affect such RNAs but not reporters with deleted start AUG. Thus, they conclude that ribosome binding is required for miRNA-mediated mRNA decay.

This is a solid statistical analysis of the impact of miRNA binding to coding and non-coding transcripts. The study addresses an important problem since it is unclear why many miRNAs associate with polysomes, for example. The finding that translation is actually required for mRNA decay is relevant and interesting. However, although the authors use reporter systems, the study would certainly benefit from more mechanistic experimental work linking ribosome binding to decay of the mRNA. There are a few comments listed below that need to be addressed.

1. The manuscript is mainly based on statistical analysis of RNA-seq, CLIP or ribosome profiling data sets. Presenting such data can be very demanding but the authors present their correlations in a clear and very nice way. However, some aspects could be clearer. On page 7, first paragraph of the part describing Figure 2 is not really accessible and unclear what exactly has been analyzed. Second, the description of Figure 4 on top of page 10 is somewhat confusing. It is stated that levels of Cdkn1a^ΔATG decrease in miRNA-depleted mESCs. Maybe one should add: ...compared to Cdkn1a wt, which is increased with miRNAs are depleted?
2. There are a number of controls that should be included. First, at least one Northern blot against a miRNA in Dicer-reduced cells should be added. In addition, when the start codon of Cdkn1a is deleted, protein levels should be assessed by western blotting or ribosome binding should be tested. In some cases, alternative AUG can be used or ribosomes can still bind through uORFs, for example.
3. As mentioned above, the manuscript would benefit from more mechanistic work explaining why translation/ribosome binding is required for miRNA-guided decay. For example, is translation of the ORF or simply recruitment of ribosomes required? To study this, the stop codon could be systematically moved towards the start codon to define a minimally required ORF. Or, it could also be interesting to investigate poly(A) tail lengths. Are poly(A) tails shortened but remain stable in the absence of ribosomes or translation? Or is only miRISC bound and deadenylases are recruited via ribosome interactions? This could be tested and would add further mechanistic insights.

Referee #2:

MicroRNAs repress protein synthesis by inhibiting mRNA translation and/or initiating mRNA deadenylation, decapping and 5'-3' destabilization. Here, Biasini and colleagues investigate whether mRNA translation is required in order for the miRNAs to promote mRNA decay. As the authors state, "a number of experiments using reporter constructs, revealed that transcript decay occurs even when translation initiation or elongation are impaired". However, they wished to address this by examining endogenous RNAs rather than reporters. To this end, they investigated long noncoding RNAs (lncRNAs) that their other data suggest interact with Argonaute proteins (CLASH and RIP assays). Their genome-wide analysis suggest that lncRNAs that are targeted by miRNAs are only minimally impacted at the level of mRNA stability as compared to protein-coding mRNAs. They go on to fuse a miRNA-targeted lncRNA (lncRNA-c1) to a GFP reporter and observe that its steady state levels are higher in DICER KO as compared to WT mES cells. Based on these data they conclude that translation is a prerequisite in order for microRNA to promote the decay of targeted RNA molecules. The paper is well-written for the most part; however, several points dampen my overall enthusiasm with respect to whether the data provided supports the main conclusions of the paper:

1. Much of the data is correlative, without a large degree of functional testing. Just because these lncRNAs are associated to a certain degree with Argonaute proteins by CLASH analysis and Argonaute RIP analysis doesn't mean that the binding sites are often bound or are even functional. Can the authors show that GW182 proteins, which are required for miRNA-mediated mRNA decay, are also associating with these lncRNAs that are not being degraded?
2. In the final figure, the authors fuse the lncRNA-c1 to the 3' of a GFP reporter and see an increase in its relative expression in DICER KO vs WT mES cells; however mutating the miRNA target sites does not seem to have a major impact on this reporter (Figure 4E, right column vs left column). Thus the effects on the construct is most likely indirect via DICER KO as opposed to directly targeting the lncRNA-c1 via a number of miRNAs. In addition, it is unclear how the authors normalized for transfection efficiency of the construct.
3. The authors mention in their discussion that their observations "are surprising in light of previous analysis demonstrating efficient miRNA dependent decay in the absence of translation initiation or elongation [23-25]. One potential confounder of previous studies is that they rely on the use of exogenous reporters". However data provided in Figure 4 uses an exogenous GFP-fused reporter and still came to different conclusions. Can the authors comment on this?
4. Ultimately, if the authors wish to assess if translation is required for the decay of a miRNA-targeted RNA, they should shut down translation using inhibitors such as hippuristanol and then look at the stability status of mRNAs that are known to be both translated and degraded.

Referee #3:

In the manuscript "Translation is required for miRNA-dependent decay of endogenous transcripts", Biasini et al. report that coding and noncoding transcripts are differentially repressed by miRNAs,

implying that translation is required for miRNA-dependent transcript destabilization. The authors compared the susceptibility of mRNAs and cytoplasmic long-non-coding RNAs (lncRNAs) to miRNA-mediated silencing in a translation-dependent manner in mouse embryonic stem cells, which revealed a correlation of translatability with effective miRNA-mediated repression. This observation was further confirmed by metabolic RNA labeling experiments in the presence and absence of Dicer to determine direct effects on RNA stability. Along those lines, putative micropeptide-encoding lncRNAs were (similar to mRNAs) more susceptible to miRNA-directed repression than untranslated lncRNAs. With the aim to experimentally validate these observations, the authors performed reporter assays to test translation-dependent miRNA-mediated repression focusing on one selected lncRNA.

While the manuscript targets an important question in the miRNA field, the presented conclusions are to a large extent based on correlative data analyses. The limited set of validation experiments (based on reporter assays) lack experimental control for translation-status and operate at a small effect size resulting in borderline statistical cutoffs. Hence, quality and impact of the manuscript would certainly benefit from more rigorous controls and expanded sample-sizes to certify the presented statistics in order to preclude any possible biases in their interpretation.

Comments:

- What miRNAs are predicted to target lncRNA-c1 and how do these compare to well-established mRNA targets that are used as a reference? Solely relying on AGO2-CLIP and AGO2-CLIP data seems quite arbitrary and non-quantitative. Is lncRNA-c1 overall (and embedded predicted miRNA target sites) conserved? This is of particular importance, given the comparison to highly responsive target mRNAs. A fairer comparison would focus on lncRNA and mRNA transcripts with equal targeting features (i.e. number of miRNA binding sites, and binding site features) and miRNA expression levels.
- The authors imply that lncRNA-c1 is not translated but there is currently no information in the manuscript as to whether it is actually depleted in ribosome profiling datasets.
- lncRNA-c1 seems to respond somewhat in steady-state abundance to DICER-depletion (Fig. S3C and D). The significance cutoff is not reported in the text. It seems that appropriate statistical evaluation would benefit from additional replicates, given that highest variance is observed among WT measurements.
- Along the previous point, translation-dependent regulation of Cdkn1a transcript was tested by comparing derepression of Cdkn1a in the presence and absence of an ATG start codon, but the effect size is small and apparently mostly driven by one outlier measurement in the case of Cdkn1a-wt. Given the rather small effect size, I would have serious concerns that the data may be overinterpreted. And even if differences in miRNA responses were to be robust across more replicates, attributing those changes to lack of translation would require confirming loss of ribosome-association in the case of delta-ATG in further control experiments, since it cannot be excluded that any downstream ATG codon could compensate.
- The X-axis values in Figure 4D are misleading and do not enable to determine the fold derepression that is observed upon miR-294 inhibitor treatment. How does this change relate to the degree of derepression observed for endogenous protein-coding mRNA targets? The effect seems to be much larger compared to the reduction in derepression observed upon mutating the predicted miR-290 target sites in GFP-lncRNA-c1-deltaMRE (Fig. 4E). How do the authors explain this?
- In the discussion the authors imply an involvement of DDX6 in discriminating translation-dependent and independent repression of miRNA targets. While I can see that those experiments may go beyond the scope of describing the phenomenon of translation-dependent miRNA-directed repression, it may add an additional layer of mechanistic understanding to the observations, which

mostly rely on correlations.

Other points:

- Most figures lack information on the number of analyzed datapoints, which makes intuitive interpretation of the data difficult.

- From the legend to Fig. 1A legend: $n = 57$ cytosolic lncRNAs and $n = 175$ nuclear lncRNAs. $57/0.066 \sim 863$ cytosolic lncRNAs; $175/0.04 = 4375$ nuclear lncRNAs.

This information contrasts with the initial 1081 cytosolic and 4953 nuclear lncRNA numbers (see p.5, line 11-12). If there is some filtering, then the criteria should be explained.

- From the legend to Fig. 1B, $n = 48$ for cytosolic lncRNA. Again, the value for the total number of cytosolic lncRNAs is different - $48/0.06 = 800$ vs 1081.

- Page 5, line 22 (and Fig. 1B): It may be helpful to show the density of Ago2 clusters for nuclear lncRNAs. As these should not be bound by miRNAs, much lower Ago2 density would speak for the correct definition of cytoplasmic and nuclear lncRNAs (definition is on p.5, lines 9-12).

- Fig2D, S2E: Interpretation would benefit from an understanding to what extent stabilized transcripts overlap between the two s4U labeling timepoints.

We would like to thank the reviewers for their insightful and constructive comments that helped us improve the manuscript and strengthen our conclusions.

In addition to addressing the reviewers' comments and to increase the robustness of our hypothesis testing we expanded our experimental analyses to an additional lncRNA candidate, *lncRNA-c2*. The results of this analysis, similarly to what we have seen for *lncRNA-c1*, support the requirement of translation for miRNA-dependent regulation. In the revised version of the manuscript, we report the results of the analysis of the two candidates in Figure 4 and Figure EV3-6 and Table EV2.

Referee #1:

Biasini et al.

Translation is required for miRNA-dependent decay of endogenous transcripts

In this study, the authors have analyzed genome-wide expression profiling data to correlate transcript levels with translation-associated, miRNA-mediated mRNA decay. Based on available data sets, the authors separated cytosolic and nuclear fractions on lncRNAs. Ribosome profiling as well as Ago2-CLIP data, which are also publicly available revealed that lncRNA rarely associated with ribosomes but do interact with miRISC. Inactivation of miRNA expression by Dicer-deletion in mESCs revealed that steady-state levels of lncRNAs are only marginally affected by miRNA expression, while mRNA levels increased as expected when Dicer is deleted. Using additional statistical tests, the authors report that miRISC binding to lncRNAs has not much impact on the expression levels. However, when they analyzed the small population of lncRNAs that contain short open reading frames and have the potential to generate micropeptides, they find that those lncRNAs are affected by the loss of miRNA expression. The authors validate their model by fusing a lncRNA to a GFP reporter and demonstrate that miRNA affect such RNAs but not reporters with deleted start AUG. Thus, they conclude that ribosome binding is required for miRNA-mediated mRNA decay.

This is a solid statistical analysis of the impact of miRNA binding to coding and non-coding transcripts. The study addresses an important problem since it is unclear why many miRNAs associate with polysomes, for example. The finding that translation is actually required for mRNA decay is relevant and interesting. However, although the authors use reporter systems, the study would certainly benefit from more mechanistic experimental work linking ribosome binding to decay of the mRNA. There are a few comments listed below that need to be addressed.

1. The manuscript is mainly based on statistical analysis of RNA-seq, CLIP or ribosome profiling data sets. Presenting such data can be very demanding but the authors present their correlations in a clear and very nice way. However, some aspects could be clearer. On page 7, first paragraph of the part describing Figure 2 is not really accessible and unclear what exactly has been analyzed. Second, the description of Figure 4 on top of page 10 is somewhat confusing. It is stated that levels of Cdkn1aATG decrease in miRNA-depleted mESCs. Maybe one should add: ...compared to Cdkn1a wt, which is increased with miRNAs are depleted?

We would like to thank the reviewer for the positive feedback. We clarified the description of Figure 2 (first paragraph page 7) as follow:

Results section:

“Of the three processes (transcription, processing and degradation) that determine steady-state transcript abundance, only the rate of degradation is directly influenced by miRNAs. To determine transcriptome-wide differences in degradation rate between miRNA depleted and wild-type mESCs, we performed, in duplicate, 4-thio-uridine (4sU, 200 μ M) metabolic labelling of RNA for 10 and 15 minutes, as previously described [1] 8 days after induction of DICER loss of function and in uninduced control mESCs. We sequenced pre-existing and newly synthesized RNA and quantified intron and exon expression transcriptome-wide in both RNA fractions (Figure 2A, Methods). Principal component analysis revealed that the RNA fraction (pre-existing or newly synthesized) is the strongest contributor to differences in gene expression between samples (Figure EV2A). The second principal component is strongly correlated with miRNA content in the cell as a result of DICER presence or absence (Figure EV2B). Finally, the third principal component discriminates samples from different biological replicates (Figure EV2C). We estimated degradation rates from the data we obtained from the two pulse durations (10 and 15 minutes) separately using INSPEcT ([2], Methods). We found that these rates are highly correlated between the two pulse lengths for wild-type and miRNA depleted cells ($R^2 > 0.75$, Figure 2B-C). As a control, we used an alternative method (transcription block by Actinomycin-D) to validate the estimated differences in transcript stability between wild-type and miRNA depleted cells for a subset of transcripts spanning a range of fold-differences in degradation rates (Pearson $R^2 = 0.58$, Figure EV2D).”

Regarding the description of Figure 4, we replaced the analysis of the *Cdkn1a* start codon mutation by a better controlled experiment and have entirely rewrote this section. We provide details on the motivation for this modification in our response to the reviewer’s point 2.

2. There are a number of controls that should be included. First, at least one Northern blot against a miRNA in Dicer-reduced cells should be added (to be cont).

The consequences on miRNA levels of conditional DICER loss of function in DT23/49 XY mESCs, the cell model used here, were previously investigated in detail in earlier publications (for example [3]; or [4]). We used small RNA-sequencing (smRNAseq) to validate these earlier observations and establish the temporal dynamics of miRNA depletion following induction of DICER loss of function (Figure 1C). This method allows sensitive and specific quantification of miRNA levels [5]. Library preparation in general and adapter ligation, in particular, can nevertheless impact the accuracy of absolute miRNA copy number estimates [6].

Despite this limitation being less of an issue when looking at relative expression levels, as done in the present study, we also sought to use an alternative approach to validate the observed decrease in miRNA levels. We did this using the

Thermofisher Advanced Taqman miRNA detection workflow, which ensures higher specificity in contrast to the conventional SYBR Green workflow. This is well illustrated by the absence of cross-reactivity between probes for human let-7 family members that differ by 1 nucleotide (<https://www.thermofisher.com/content/dam/LifeTech/Documents/PDFs/TaqMan-Advanced-miRNA-Performance-White-Paper.pdf>). Our analysis of two of the most abundant miRNAs in mESCs (reported in Figure EV1) confirmed that consistent with our smRNAseq experiment, 8 days after DICER loss of function, the levels of these two abundant miRNAs are at least 90% lower than in wild-type cells.

(cont) In addition, when the start codon of Cdkn1a is deleted, protein levels should be assessed by western blotting or ribosome binding should be tested. In some cases, alternative AUG can be used or ribosomes can still bind through uORFs, for example.

We share the reviewer's concerns regarding the consequences of canonical start codon mutations on Cdkn1a translation. To overcome the potential shortcomings of this approach, that include for example initiation of translation from an alternative start codon that are hard to quantify, we decided to analyze the impact of inserting 5 BoxB sites upstream of the start codon in GFP-IncRNA candidate constructs as previously done [7]. Addition of these hairpin structures results in a significant reduction in GFP levels measured by FACS and Western Blot (>25-fold, Figure 4 and Figure EV5).

Figure 4F-Representative immunoblot analysis of GFP (GFP) in protein extracts from mESCs transfected with mock, BoxB(-30)-GFP and GFP expressing vectors. ACTIN- β (ACT- β) was used as an internal control. Uncropped blots used for assembly of panels F is provided in Figure 4 Source Data.

Using these constructs, that we named BoxB(-30)-GFP-IncRNA-c1/c2, we show that impaired GFPIncRNA-c1 or GFPIncRNA-c2 translations is associated with a significant decrease in the impact of miRNA on reporter gene expression (Figure 4).

Figure 4G- Fold-change (FC) in normalized expression of GFP-IncRNA-c1, GFP-IncRNA-c2 (noBoxB, circles) and BoxB(-30)-GFP-IncRNA-c1, BoxB(-30)-GFP-IncRNA-c2 (BoxB(-30), triangles) (x-axis) 4-OHT treated, miRNA depleted mESCs (KO) relative to ethanol treated mESCs (WT) (y-axis). Four independent biological replicates were analyzed For all RTqPCR analyses, transcript expression was first normalized by the amount of *Act-β* and *PoIII* and next by the total amount of transfected vectors per cell estimated based on the levels of relative *Neomycin* expression. Each point corresponds to the results of one independent biological replicate. Statistics: *-p-value<0.05, **-p-value<0.01 and ***-p-value<0.001.

3. As mentioned above, the manuscript would benefit from more mechanistical work explaining why translation/ribosome binding is required for miRNA-guided decay. For example, is translation of the ORF or simply recruitment of ribosomes required? To study this, the stop codon could be systematically moved towards the start codon to define a minimally required ORF. (to be cont)

To understand whether translation of the ORF or simply recruitment of ribosomes was required in addition to BoxB(-30)GFP-IncRNA-c1/c2 we also analyzed how inhibiting translation elongation, by inserting 5xBoxB 339 nucleotides upstream of GFP start codon (BoxB(+339)GFP- IncRNA-c1/c2). We found that introduction of these hairpin structures decreased GFP levels supporting that this modification leads to inhibition of translation elongation (Figure 4H).

Figure 4H-Immunoblot analysis of GFP (GFP) in protein extracts from mESCs transfected with mock, BoxB(+339)-GFP and GFP expressing vectors. ACTIN-β (ACT-β) was used as an internal control. One representative blot is depicted. Uncropped blots used for assembly of panel H is provided in Figure 4 Source Data.

Importantly, we found that BoxB(+339) has a similar impact on the response of GFP-IncRNA candidate to miRNA depletion (Figure 4I). These results suggest that translation elongation is required for miRNA dependent degradation.

Figure 4I- Fold-change (FC) in normalized expression of GFP-lncRNA-c1, GFP-lncRNA-c2 (noBoxB, circles) and BoxB(+339)-GFP-lncRNA-c1, BoxB(+339)-GFP-lncRNA-c2 (BoxB(+339), triangles) (x-axis) in 8 day 4-OHT treated, miRNA depleted mESCs (KO) relative to ethanol treated mESCs (WT) (y-axis). Four independent biological replicates analyzed. For all RTqPCR analyses, transcript expression was first normalized by the amount of *Act-β* and *PolIII* and next by the total amount of transfected vectors per cell estimated based on the levels of relative *Neomycin* expression. Each point corresponds to the results of one independent biological replicate. Statistics: *-p-value<0.05, **-p-value<0.01 and ***-p-value<0.001.

(cont.) Or, it could also be interesting to investigate poly(A) tail lengths. Are poly(A) tails shortened but remain stable in the absence of ribosomes or translation? Or is only miRISC bound and deadenylases are recruited via ribosome interactions? This could be tested and would add further mechanistic insights.

We attempted to investigate the impact of miRNA depletion on coding and noncoding transcript polyA length. For this we chose to use the Poly(A) Tail-Length Assay Kit by Thermo Fisher which did not require establishing and ordering radioactivity reagents which would have not been possible due to COVID working restrictions. Whereas analysis of the positive control indicated that the chemistry worked in our hands, we were unfortunately unable because of the repetitive nature of the lncRNAs 3'end to design a primer that would allow specific amplification of lncRNAc1 or c2 polyA tail. The alternative would be to study the impact of miRNA depletion on coding and noncoding transcript polyA length genome wide which is a relevant follow-up as we now mention in the discussion.

Discussion Section:

“More generally the present results also imply that miRISC binding, per se, is not sufficient to determine the outcome of bound targets suggesting the requirement of further yet unidentified molecular partners. For example, GW182, a major component of miRISC, recruits the CCR4-NOT deadenylase complex [8, 9]. Deadenylation of mRNAs by CCR4-NOT has been shown to be a key step in facilitating miRNA-mediated transcript decay [10, 11]. Understanding the differences in coding and noncoding RNA miRNA-dependent deadenylation, for example by investigating polyA length following DICER depletion genome wide or identifying translation-dependent factors, may provide the much-needed mechanistic insights on how translation facilitates miRNA-dependent gene regulation.”

Referee #2:

MicroRNAs repress protein synthesis by inhibiting mRNA translation and/or initiating mRNA deadenylation, decapping and 5'-3' destabilization. Here, Biasini and colleagues investigate whether mRNA translation is required in order for the miRNAs to promote mRNA decay. As the authors state, "a number of experiments using reporter constructs, revealed that transcript decay occurs even when translation initiation or elongation are impaired". However, they wished to address this by examining endogenous RNAs rather than reporters. To this end, they investigated long noncoding RNAs (lncRNAs) that their other data suggest interact with Argonaute proteins (CLASH and RIP assays). Their genome-wide analysis suggest that lncRNAs that are targeted by miRNAs are only minimally impacted at the level of mRNA stability as compared to protein-coding mRNAs. They go on to fuse a miRNA-targeted lncRNA (lncRNA-c1) to a GFP reporter and observe that its steady state levels are higher in DICER KO as compared to WT mES cells. Based on these data they conclude that translation is a prerequisite in order for microRNA to promote the decay of targeted RNA molecules. The paper is well-written for the most part; however, several points dampen my overall enthusiasm with respect to whether the data provided supports the main conclusions of the paper:

1. Much of the data is correlative, without a large degree of functional testing. Just because these lncRNAs are associated to a certain degree with Argonaute proteins by CLASH analysis and Argonaute RIP analysis doesn't mean that the binding sites are often bound or are even functional. Can the authors show that GW182 proteins, which are required for miRNA-mediated mRNA decay, are also associating with these lncRNAs that are not being degraded?

In addition to the genome-wide analysis of AGO2 association to lncRNAs (Figure 1C) and evidence that transcripts bound by AGO2 are more responsive to miRNA depletion (Figure 2E), we validated the functionality of the predicted miRNA binding sites using miRNA mimics and inhibitors for lncRNA-c1 (Figure 4C) and Extended View File 6C-F.

The development of Halo-enhanced Ago2 pull-down (HEAP) [12] and the recent release of AGO2 bound sites in mESCs determined using this approach, allowed us to further validate the interaction between miRISC and cytosolic lncRNAs, in general, and our candidate lncRNAs, in particular. Given this data extends significantly on the number of mESC transcripts bound by AGO2, we decided to replace the analysis done previously with AGO2-CLIP data with HEAP. The AGO2-CLIP results, that are concordant with the ones obtained using HEAP, are now presented in Extended View File 1C. which supports the functionality of MRE in candidate lncRNAs

To complement these analyses, we attempted to perform a GW182-RIP in mESCs, as suggested by the reviewer, using the TNRC6A polyclonal antibody from Abnova. This is a synthetic peptide corresponding to the 16aa near C-Terminus of Human TNRC6A. In our hands, this antibody can be used to successfully IP TNRC6A in HEK293 cells. Unfortunately, and as illustrated by the Western Blot below, the antibody does not react with the mouse TNRC6A despite the vendor's claims. None of the bands we obtained has the size expected for TNRC6A (182kDa) and the non-specific band we observed at around 100kDa suggests that the antibody is not

suitable for this application. After searching different vendor catalogs we could not identify a mouse antibody that was validated for IP experiments in mouse. We could not find papers reporting on GW182 -RIP/CLIP in mouse that would guide the choice of an alternative antibody. So, despite our efforts we could not do the experiment requested by this reviewer.

Figure Reviewer2: Immunoprecipitation of TNRC6A in E14 mESC line.

2. In the final figure, the authors fuse the lincRNA-c1 to the 3' of a GFP reporter and see an increase in its relative expression in DICER KO vs WT mES cells; however mutating the miRNA target sites does not seem to have a major impact on this reporter (Figure 4E, right column vs left column). Thus the effects on the construct is most likely indirect via DICER KO as opposed to directly targeting the lincRNA-c1 via a number of miRNAs. In addition, it is unclear how the authors normalized for transfection efficiency of the construct.

To account for differences in transfection efficiency between experiments and cell types, we normalize reporter expression (relative to *Actin-β* and *PolymeraseII*) to the expression of Neomycin. Neomycin is transcribed from an independent promoter in the same vector. This is mentioned in the figure legends and detailed in the methods section.

“Gene expression levels relative to *Actin-β* and *Polymerase II* of transfected candidates were normalized to *Neomycin* expression to account for differences in transfection efficiency between different cell types and experiments”.

Whereas we agree with the reviewer that the MRE mutations do not fully rescue the effects of miRNAs on *GFP-IncRNA-c1* expression, we disagree that this is an indication that the observed differences are likely an indirect effect of DICER loss of function. We attribute the reviewer’s comment to the way the data was depicted in the previous version of the manuscript and that we have now modified.

Figure 4C- Expression of *GFP-IncRNA-c1* Δ MRE relative to *GFP-IncRNA-c1* (y-axis) in ethanol treated (WT, circles) or 4-OHT treated, miRNA-depleted cells (KO, triangles, x-axis). Four independent biological replicates were treated, transfected and analyzed by RTqPCR. For all RTqPCR analyses, transcript expression was first normalized by the amount of *Act-β* and *PoIII* and next by the total amount of transfected vectors per cell estimated based on the levels of relative *Neomycin* expression. Each point corresponds to the results of one independent biological replicate. Statistics: NS- p-value > 0.05, *-p-value < 0.05, **-p-value < 0.01 and ***-p-value < 0.001. Uncropped blots used for assembly of panels F and H are provided in Figure 4 Source Data.

As can be appreciated in the modified version Figure 4D, MRE deletion in *IncRNA-c1* is associated with increased abundance of reporter construct in WT cells relative to reporter gene with intact MREs. Importantly the increase we observe in *GFP-IncRNA-c1* Δ MRE expression relative to *GFP-IncRNA-c1* is significantly larger in WT cells. This observation supports that MRE deletion primarily affects the levels of reporter genes in WT cells, as expected. We attribute the incomplete rescue of MRE mutations, to the presence of functional binding sites for other miRNAs. As can be appreciated in Extended View File 8 we mutated 3 of the 17 MREs in *IncRNA-c1* for mESC expressed miRNAs, respectively. Whereas we chose to focus on MREs for the most highly expressed miRNAs in mESCs [13] a number of other less abundant miRNAs in these cells have predicted MREs and can also impact the levels of targets. We modified the following section to clarify this point.

Results section:

“Evidence of AGO2-binding to one of *IncRNA-c1*’s miR-290/295 family of miRNAs response elements (MREs) supports that this candidate is bound by AGO2 loaded with one or more members of this miRNA family (Figure EV5A-B and Figure EV6). To validate that these miRNAs are indeed contributing to miRNA-dependent repression of *GFP-IncRNA-c1*, we co-transfected mESCs with *GFP-IncRNA-c1* expressing vector and miR-294-inhibitors. We note a significantly higher expression

of GFP-*lncRNA-c1* in the inhibitor transfected cells compared to cells transfected with negative control (paired two-tailed t-test p -value <0.006 , Figure EV5C). We used site-directed mutagenesis to mutate three MREs for members of the miR-290/295 highly expressed miRNA family within GFP-*lncRNA-c1* (hereafter GFP-*lncRNA-c1* Δ MRE). As expected, reintroduction of miRNA mimics in DICER depleted mESC impacts significantly the levels of these miRNA endogenous targets, *Cdkn1a* (paired two-tailed t-test p -value=0.024, Figure EV5D) and *Lats2* (paired two-tailed t-test p -value=0.003, Figure EV5E). Consistent with the functionality of miR-290/295 MREs within *lncRNA-c1*, GFP-*lncRNA-c1* is significantly more downregulated upon miRNA mimic reintroduction than GFP-*lncRNA-c1* Δ MRE (paired one-tailed t-test p -value=0.043, Figure EV5F). Consistent with these MREs mediating in part wild-type GFP-*lncRNA-c1* miRNA-dependent repression, we found that disrupting miRNA binding sites attenuated the impact of miRNA depletion on GFP-*lncRNA-c1* levels (paired two-tailed t-test p -value=0.007, Figure 4C). We noted that GFP-*lncRNA-c1* is more responsive to global miRNA depletion than to MRE mutation. The presence of additional MREs for other mESC-expressed miRNAs (14, Table EV2) likely explains why mutation of highly expressed miR290/295 seed-complementary MREs is not sufficient to entirely abolish miRNA-dependent GFP-*lncRNA-c1* destabilization.”

3. The authors mention in their discussion that their observations "are surprising in light of previous analysis demonstrating efficient miRNA dependent decay in the absence of translation initiation or elongation [23-25]. One potential confounder of previous studies is that they rely on the use of exogenous reporters". However, data provided in Figure 4 uses an exogenous GFP-fused reporter and still came to different conclusions. Can the authors comment on this?

Our conclusions are based on the genome-wide analysis of endogenously expressed transcripts that we went on to validate using a series of reporter constructs. The main difference between our reporter analysis and what was done in previous studies is that we seldomly rely on miRNA mimic/inhibitor transfections. The use of miRNA mimic/inhibitors is generally associated with a strong impact of miRNA on target expression as we can also appreciate in our experiments (Figure EV5C). In contrast, the impact of miRNA depletion on endogenous and exogenous transcript expression we observe is in line with the reported physiological impact of miRNA on target expression [14, 15] and is relatively modest (Figure 4B). Our hypothesis is that the non-physiological responses to mimics/inhibitor induced miRNA levels likely masks the contribution of translation to miRNA dependent regulation. This is supported by evidence that despite these limitations, a decrease in miRNA dependent regulation is still observed when translation is inhibited in these studies [7, 16]. We clarified this point in the discussion.

Discussion section:

“These observations may appear surprising in light of previous reports that miRNA dependent decay is minimally impacted by translation initiation or elongation [7, 17, 18]. One potential confounder of previous studies is that they rely on the use of exogenous reporters. Furthermore, the level of gene expression changes induced by the use of mimics or inhibitors exceed by several fold the expected impact of miRNAs on their target levels [14, 15]. Together these and other factors may limit the

extent by which these earlier studies recapitulate what happens *in vivo*. Analysis of cytosolic bona fide lncRNAs, that have been previously shown to interact with miRISC [19] but not with the translation machinery [20], provide a unique opportunity to investigate the requirement of translation for endogenous miRNA-directed target decay. Furthermore, the relatively modest impact of miRNA depletion on target levels we observed following inducible DICER loss of function is in line with what would be expected for physiological relevant miRNA-target interactions [14, 15], supporting the use of this experimental system and the physiological relevance of our findings.”

4. Ultimately, if the authors wish to assess if translation is required for the decay of a miRNA-targeted RNA, they should shut down translation using inhibitors such as hippuristanol and then look at the stability status of mRNAs that are known to be both translated and degraded.

To address this and the other reviewers concerns we decided to introduce 5BoxB sites upstream of the GFP start codon [BoxB(-30)] and in the middle of GFP ORF [BoxB(+339)].

Figure EV5G- Schematics of the BoxB insertion in GFP-lncRNA-candidate constructs

Addition of these hairpin structures resulted in significant reduction in translation initiation and elongation (for example Figure 4F,H).

Figure 4F- Representative immunoblot analysis of GFP (GFP) in protein extracts from mESCs transfected with mock, BoxB(-30)-GFP and GFP expressing vectors. ACTIN-β(ACT-β) was used as an internal control. Uncropped blots used for assembly of panels F is provided in Appendix Figure 4 Source Data

Figure 4H-Immunoblot analysis of GFP (GFP) in protein extracts from mESCs transfected with mock, BoxB(+339)-GFP and GFP expressing vectors. ACTIN- β (ACT- β) was used as an internal control. One representative blot is depicted. Uncropped blots used for assembly of panel H is provided in Figure 4 Source Data.

This approach is conceptually similar to the one proposed by the reviewer but minimizes pleiotropic effects due to the use of translation inhibition drugs. We found that decreased translation inhibition or elongation of GFPIncRNA-c1 and GFPIncRNA-c2 (Figure 4G and I) is associated with a significant decrease in the impact of miRNA on reporter gene expression.

Figure 4G- Fold-change (FC) in normalized expression of *GFP-IncRNA-c1*, *GFP-IncRNA-c2* (noBoxB, circles) and *BoxB(-30)-GFP-IncRNA-c1*, *BoxB(-30)-GFP-IncRNA-c2* (BoxB(-30), triangles) (x-axis) 4-OHT treated, miRNA depleted mESCs (KO) relative to ethanol treated mESCs (WT) (y-axis). Four independent biological replicates analyzed. For all RTqPCR analyses, transcript expression was first normalized by the amount of *Act- β* and *PoIII* and next by the total amount of transfected vectors per cell estimated based on the levels of relative *Neomycin* expression. Each point corresponds to the results of one independent biological replicate. Statistics: *-p-value<0.05, **-p-value<0.01 and ***-p-value<0.001

Figure 4I—Fold-change (FC) in normalized expression of *GFP-IncRNA-c1*, *GFP-IncRNA-c2* (noBoxB, circles) and *BoxB(+339)-GFP-IncRNA-c1*, *BoxB(+339)-GFP-IncRNA-c2* (BoxB(+339), triangles) (x-axis) in 4-OHT treated, miRNA depleted mESCs (KO) relative to ethanol treated mESCs (WT) (y-axis). Four independent biological replicates analyzed. For all RTqPCR analyses, transcript expression was first normalized by the amount of *Actin-β* and *PolymeraseII* and next by the total amount of transfected vectors per cell estimated based on the levels of relative *Neomycin* expression. Each point corresponds to the results of one independent biological replicate. Statistics: *-p-value<0.05, **-p-value<0.01 and ***-p-value<0.001

Referee #3:

In the manuscript "Translation is required for miRNA-dependent decay of endogenous transcripts", Biasini et al. report that coding and noncoding transcripts are differentially repressed by miRNAs, implying that translation is required for miRNA-dependent transcript destabilization. The authors compared the susceptibility of mRNAs and cytoplasmic long-non-coding RNAs (lncRNAs) to miRNA-mediated silencing in a translation-dependent manner in mouse embryonic stem cells, which revealed a correlation of translatability with effective miRNA-mediated repression. This observation was further confirmed by metabolic RNA labeling experiments in the presence and absence of Dicer to determine direct effects on RNA stability. Along those lines, putative micropeptide-encoding lncRNAs were (similar to mRNAs) more susceptible to miRNA-directed repression than untranslated lncRNAs. With the aim to experimentally validate these observations, the authors performed reporter assays to test translation-dependent miRNA-mediated repression focusing on one selected lncRNA.

While the manuscript targets an important question in the miRNA field, the presented conclusions are to a large extent based on correlative data analyses. The limited set of validation experiments (based on reporter assays) lack experimental control for translation-status and operate at a small effect size resulting in borderline statistical cutoffs. Hence, quality and impact of the manuscript would certainly benefit from more rigorous controls and expanded sample-sizes to certify the presented statistics in order to preclude any possible biases in their interpretation.

Comments:

1- What miRNAs are predicted to target *lncRNA-c1* and how do these compare to well-established mRNA targets that are used as a reference? Solely relying on AGO2-CLIP and AGO2-CLIP data seems quite arbitrary and non-quantitative. Is *lncRNA-c1* overall (and embedded predicted miRNA target sites) conserved? This is of particular importance, given the comparison to highly responsive target mRNAs. A fairer comparison would focus on *lncRNA* and mRNA transcripts with equal targeting features (i.e. number of miRNA binding sites, and binding site features) and miRNA expression levels.

The list of mESC expressed miRNAs that are predicted, by TargetScan, to target *lncRNA-c1* is provided in Extended View File 8. Besides miRNA predictions, in the previous version of the manuscript we used publicly available CLIP data [21] to assess the extent AGO2 bound *lncRNAs* genome wide (Figure 1B and Extended View File 1C). In the revised version of the manuscript, we repeated this analysis using the recently released HEAP data in mESCs [12] The results of this analysis are consistent with those presented for AGO2-CLIP [21] and support binding of cytosolic *lncRNAs* in general (Figure 1C) and candidate *lncRNA* (Figure EV5A) in particular.

Figure 1B - Density of HEAP-AGO2 peaks across cytosolic *lncRNAs* (n=62, blue) and the 3'UTR regions of mRNAs (n=8798, red) with experimental evidence for AGO2 binding in mESCs (>0 AGO2 peaks).

Figure EV3A Genome browser view of the region encompassing *lncRNA-c1*, Halo enhanced AGO2 pull-down peaks [12] and read density for two independent replicates (between 0-127) as well as PhyloCSF scores (between -15 and 15) in all possible reading frames are depicted. Gencode annotated genes are annotated in blue and the candidates are annotated in black.

We agree with the reviewer that comparison between lncRNA and *Cdkn1a* may be potentially confounded by differences in MRE responsiveness. To account for this confounder as well as other limitations of the previous experiment (detailed in our response to Reviewer 1, point 2 and this reviewer point 3) in the revised version of the manuscript, we compared how inhibiting translation of reporter gene by inserting a 5 BoxB sites upstream of start codon impacts reporter's response to miRNAs depletion. Given that these reporters only differ at their 5'UTR (Extended View File 6D), this is a fairer comparison than the one we presented in the previous version of the manuscript where we compared lncRNAs with and a highly responsive mRNA target, *Cdkn1a*. The results of this experiment, that we describe our response to this reviewer point 4, are consistent with the requirement of translation to miRNA dependent regulation

Figure EV5G- Schematics of the BoxB insertion in GFP-lncRNA-candidate constructs

In addition, and as suggested by the reviewer, we also add the alignment of the MRE regions in *lncRNA-c1* as Extended View File 7. This analysis revealed that the sequence of *lncRNA-c1* is partially conserved between mouse, rat and human And, that only one the MREs we mutated within *lncRNA-c1* is conserved in all the species considered.

Figure EV6- Alignment in mouse, rat and human of the mutated MRE sequences in *lncRNA-c1*. Seed region is highlighted in grey. = represent positions not conserved and – deletions in either rat or human. Deletions in mouse are represented in yellow and the length indicated in the alignment.

2 The authors imply that lncRNA-c1 is not translated but there is currently no information in the manuscript as to whether it is actually depleted in ribosome profiling datasets.

We would like to thank the reviewer for pointing this out. We added the nucleotide PhyloCSF score for lncRNA-c1 and c2, that reflects the likelihood a position encodes a conserved ORF to the revised version of the Extended View File 4A-B and added a figure depicting the density of Ribosomal Profiling reads in mESCs for the two candidate lncRNAs in Extended View File 5. Based on this, we conclude the two lncRNA candidates we selected are unlikely to encode for a protein. We describe this in the result section.

“To test this hypothesis, we selected two cytosolic lncRNAs expressed at relatively different levels in mESCs: TCONS_00034281 and TCONS_00031378, hereafter lncRNA-c1 and lncRNA-c2 respectively (Figure EV3A-C). Both lncRNAs lack an apparent conserved ORF (Figure EV3A-B) and are weakly associated with ribosomes (Figure EV4A-B) supporting that they are noncoding.”

3 lncRNA-c1 seems to respond somewhat in steady-state abundance to DICER-depletion (Fig. S3C and D). The significance cutoff is not reported in the text. It seems that appropriate statistical evaluation would benefit from additional replicates, given that highest variance is observed among WT measurements.

We think the difference in variance the reviewer points out is a technical artifact. We see similar variability in lncRNA-c1 levels in WT and KO samples when expression was measured by RT-qPCR using a completely independent set of samples. We added the p-values to the main text as requested by the reviewer.

Results section:

“Our transcriptome wide analysis indicates that translation is required for miRNA-dependent target destabilization. To test this hypothesis, we selected two cytosolic lncRNAs expressed at relatively different levels in mESCs: TCONS_00034281 and TCONS_00031378, hereafter lncRNA-c1 and lncRNA-c2 respectively (Extended View File 4A-C). Both lncRNAs lack an apparent conserved ORF (Extended View File 4A-B) and are weakly associated with ribosomes (Extended View File 5) supporting that they are noncoding. Consistent with our transcriptome wide profiling experiment (paired two-tailed t-test p-value >0.2, Extended View File 4D), RT-qPCR analysis also support that the steady state abundance of endogenously expressed lncRNA-c1 and lncRNA-c2 are not significantly increased upon miRNA depletion (paired two-tailed t-test p-value >0.5, Extended View File 4E), in line with our hypothesis and in contrast to bonafide miRNA targets, such as Lats2 or Cdkn1A [22] (paired two-tailed t-test p-value < 0.04, Extended View File 4D,E). Furthermore, and in contrast with Lats2 or Cdkn1A (paired two-tailed t-test p-value < 0.03, Extended View File 4F), stability of lncRNA-c1 and lncRNA-c2 is also not significantly affected in cells lacking DICER function (paired two-tailed t-test p-value > 0.2, Extended View File 4F)”

4 Along the previous point, translation-dependent regulation of Cdkn1a transcript was tested by comparing derepression of Cdkn1a in the presence and absence of an ATG start codon, but the effect size is small and apparently mostly driven by one outlier measurement in the case of Cdkn1a-wt. Given the rather small effect size, I would have serious concerns that the data may be overinterpreted. And even if differences in miRNA responses were to be robust across more replicates, attributing those changes to lack of translation would require confirming loss of ribosome-association in the case of delta-ATG in further control experiments, since it cannot be excluded that any downstream ATG codon could compensate.

We share the reviewer's concerns regarding the impact of canonical start site mutations on Cdkn1a translation. Specifically, the experiment presented in the previous version of the manuscript did not account for potential initiation of translation downstream of mutated start codon (this point) as well as potential differences in the responsiveness to miRNAs (point1) as pointed out by the reviewer. To overcome these limitations, we chose to focus our analysis on the set of reporters that contain a 5 BoxB sites either 30 or 339 nucleotides down- or upstream of GFP start codon.

Figure EV5G- Schematics of the BoxB insertion in GFP-lncRNA-candidate constructs

Addition of these hairpin structures resulted in significant reduction in translation (for example Figure 4F-H).

Figure 4F- Representative immunoblot analysis of GFP (GFP) in protein extracts from mESCs transfected with mock, BoxB(-30)-GFP and GFP expressing vectors. ACTIN- β (ACT- β) was used as an internal control. Uncropped blots used for assembly of panels F is provided in Figure 4 Source Data

Figure 4H-Immunoblot analysis of GFP (GFP) in protein extracts from mESCs transfected with mock, BoxB(+339)-GFP and GFP expressing vectors. ACTIN- β (ACT- β) was used as an internal control. One representative blot is depicted. Uncropped blots used for assembly of panel H is provided in Figure 4 Source Data.

We found that inhibiting translation initiation or elongation of GFPIncrRNA-c1 and GFPIncrRNA-c2 (Figure 4G and I) is associated with a significant decrease in the impact of miRNA on reporter gene expression.

Figure 4G- Fold-change (FC) in normalized expression of *GFP-IncrRNA-c1*, *GFP-IncrRNA-c2* (noBoxB, circles) and *BoxB(-30)-GFP-IncrRNA-c1*, *BoxB(-30)-GFP-IncrRNA-c2* (BoxB(-30), triangles) (x-axis) 4-OHT treated, miRNA depleted mESCs (KO) relative to ethanol treated mESCs (WT) (y-axis). Four independent biological replicates analyzed. For all RTqPCR analyses, transcript expression was first normalized by the amount of *Actin- β* and *PolymeraseII* and next by the total amount of transfected vectors per cell estimated based on the levels of relative *Neomycin* expression. Each point corresponds to the results of one independent biological replicate. Statistics: *-p-value<0.05, **-p-value<0.01 and ***-p-value<0.001.

Figure 4I-Fold-change (FC) in normalized expression of *GFP-IncRNA-c1*, *GFP-IncRNA-c2* (noBoxB, circles) and *BoxB(+339)-GFP-IncRNA-c1*, *BoxB(+339)-GFP-IncRNA-c2* (BoxB(+339), triangles) (x-axis) in 4-OHT treated, miRNA depleted mESCs (KO) relative to ethanol treated mESCs (WT) (y-axis). Four independent biological replicates analyzed. For all RTqPCR analyses, transcript expression was first normalized by the amount of *Actin-β* and *PolymeraseII* and next by the total amount of transfected vectors per cell estimated based on the levels of relative *Neomycin* expression. Each point corresponds to the results of one independent biological replicate. Statistics: *-p-value<0.05, **-p-value<0.01 and ***-p-value<0.001.

- The X-axis values in Figure 4D are misleading and do not enable to determine the fold derepression that is observed upon miR-294 inhibitor treatment. How does this change relate to the degree of derepression observed for endogenous protein-coding mRNA targets? The effect seems to be much larger compared to the reduction in derepression observed upon mutating the predicted miR-290 target sites in *GFP-IncRNA-c1-deltaMRE* (Fig.4E). How do the authors explain this?

We agree with the reviewer that the inhibitor experiment would benefit from the addition of internal controls. However, we had no sample left from this experiment and could not do this analysis.

Furthermore, the experiment that, in our opinion, is key to demonstrate the functional relevance of the mutated MREs is the one we report in EVF6F. In this experiment, we tested how miRNA reintroduction in DICER depleted mESCs impacted the levels of *GFP-IncRNA-c1* and *GFP-IncRNA-c1ΔMRE*.

Figure EV5F- Fold change in expression (y-axis) of *GFP*, *GFP-IncRNA-c1* and *GFP-IncRNA-c1-MREΔ* (x-axis) in miRNA depleted cells transfected with negative control (NC) relative to miRNA depleted cells transfected with miRNA mimics (miRNA) (y-axis). Each point corresponds to the results of one independent biological replicate. Transcript expression was first normalized by the amount of

Actin-b and *Polymerasell* and next by the total amount of transfected vectors per cell estimated based on levels of relative *Neomycin* expression. Statistics: *-p-value<0.05, **-p-value<0.01 and ***-p-value<0.001.

We now show that the levels of endogenous miRNA targets are also affected in this experiment.

Figure EV5D and E - Relative *Cdkn1a* (D) and *Lats2* (E) expression following transfection of mmu-miR-294-3p and mmu-miR-295-3p equimolar mixes (Mimic) or negative control small RNA (NC). Each point corresponds to the results of one independent biological replicate. Transcript expression was first normalized by the amount of *Actin-β* and *Polymerasell*. Each point corresponds to the results of one biological replicate. Statistics: *-p-value<0.05, **-p-value<0.01 and ***-p-value<0.001.

Regarding the difference in the effect observed using miRNA inhibitors relative to MRE mutation. Treatment with miR-294 antimir impacts GFP-lncRNA abundance directly because it reduces the levels of miRNAs available for GFP-lncRNA regulation. This direct effect is similar to disrupting MREs within GFP-lncRNAs. However, unlike MRE mutations, antimir will also impact the levels of other gene expression regulators targeted by miR-294 and consequently globally impact cell physiology. These pleiotropic effects, that are not present when performing MRE mutation, can also impact GFP-lncRNA levels. In addition, antimir have also been shown to associate with off-targets [23] and reduce the availability of miRISC [24]. We added this point to the results section.

- In the discussion the authors imply an involvement of DDX6 in discriminating translation-dependent and independent repression of miRNA targets. While I can see that those experiments may go beyond the scope of describing the phenomenon of translation-dependent miRNA-directed repression, it may add an additional layer of mechanistic understanding to the observations, which mostly rely on correlations.

To understand whether translation of the ORF or simply recruitment of ribosomes was required in addition to BoxB(-30)GFP-lncRNA-c1/c2 (described in our response to this reviewer's point #4) we also analyzed how inhibiting translation elongation, by inserting 5xBoxB 339 nucleotides upstream of GFP start codon. We found that

introduction of these hairpin structures decreased GFP levels supporting that this modification leads to inhibition of translation elongation. Importantly, we found that BoxB(+339) has a similar impact on the response of GFP-lncRNA candidate to miRNA depletion. These results suggest that translation elongation is required for miRNA dependent degradation.

Other points:

- Most figures lack information on the number of analyzed datapoints, which makes intuitive interpretation of the data difficult.

We have now specified in the text of the figure legends the number of analyzed data points.

- From the legend to Fig. 1A legend: $n = 57$ cytosolic lncRNAs and $n = 175$ nuclear lncRNAs. $57/0.066 \sim 863$ cytosolic lncRNAs; $175/0.04 = 4375$ nuclear lncRNAs. This information contrasts with the initial 1081 cytosolic and 4953 nuclear lncRNA numbers (see p.5, line 11-12). If there is some filtering, then the criteria should be explained.

Differences in sequencing depth between different experiments explain the differences pointed out by the reviewer. Specifically, the RNA sequencing data we used to infer lncRNA subcellular localization has over 4 times more reads than the RP data. We now clarify this in the materials and methods section of the Manuscript and would like to thank the reviewer for pointing this out.

Methods section:

“Translational efficiency (TE) was calculated as the log₂ ratio between normalized RP counts and normalized TR counts. TE was only calculated for genes with cpm>1 and have RP read>0.”

- From the legend to Fig. 1B, $n = 48$ for cytosolic lncRNA. Again, the value for the total number of cytosolic lncRNAs is different - $48/0.06 = 800$ vs 1081.

In Figure 1B we only consider transcripts with evidence for AGO2 binding. We specify this in the Results section and the figure legend. Note the numbers below differ from those presented by the reviewer because they refer to Halo enhanced AGO2 pull-down in mESCs, which we used to replace the AGO2-CLIP analysis in the main text. We still present AGO2-CLIP results as supplementary data.

Results section:

“...the density of binding at cytosolic lncRNAs (1.2 sites per kb of sequence) and mRNA 3'UTRs (1.4 sites per kb of sequence for the transcripts with experimental

evidence for AGO2 binding (>0 peak)) is similar (two-tailed Mann-Whitney U test, p -value=0.166, Figure 1B).”

Figure 1 B legend:

“Density of HEAP-AGO2 peaks across cytosolic lncRNAs ($n=62$, blue) and the 3’UTR regions of mRNAs ($n=8798$, red) with experimental evidence for AGO2 binding in mESCs (>0 AGO2 peaks).”

- Page 5, line 22 (and Fig. 1B): It may be helpful to show the density of Ago2 clusters for nuclear lncRNAs. As these should not be bound by miRNAs, much lower Ago2 density would speak for the correct definition of cytoplasmic and nuclear lncRNAs (definition is on p.5, lines 9-12).

We have now added the results of this analysis to the text. As expected a significantly higher fraction of cytosolic lncRNAs are bound by AGO2 according to this analysis.

Results section:

“As expected, nuclear lncRNAs have a significantly lower (two-tailed Mann-Whitney test, p -value<0.005) density of AGO2 peaks (0.7 sites per kb) than cytosolic lncRNAs.”

- Fig2D, S2E: Interpretation would benefit from an understanding to what extent stabilized transcripts overlap between the two s4U labeling timepoints.

The same set of transcripts were analyzed in both experiments and their degradation rates are highly correlated between the experiments as depicted in Figure 2B-C. We added the numbers of transcripts that are stabilized and significantly stabilized in both experiments to the results section.

Results section:

“As expected, miRNA depletion is associated with an increase in transcript stability (85% and 91% of transcripts have lower degradation rate in KO cells for 10 and 15 minutes pulse, Figure 2D and Figure EV2E, respectively). Next, we identified genes whose degradation rate is significantly different ($FDR<0.05$) between miRNA-depleted and control mESCs (14% and 17% in 10 and 15 minutes pulse, Figure 2D and Figure EV2E, respectively) and found that as expected, mRNAs are significantly more often stabilized in miRNA-depleted mESCs relative to control”

Figure 2B-C- Correlation between degradation rates (\log_{10}) obtained after 10 (x-axis) and 15 (y-axis) minutes of 200 μ M 4sU labelling in wildtype (WT) (**B**) and DICER null (KO) (**C**) cells.

1. Biasini A, Marques AC: **A Protocol for Transcriptome-Wide Inference of RNA Metabolic Rates in Mouse Embryonic Stem Cells.** *Front Cell Dev Biol* 2020, **8**:97.
2. de Pretis S, Kress T, Morelli MJ, Melloni GE, Riva L, Amati B, Pelizzola M: **INSPECT: a computational tool to infer mRNA synthesis, processing and degradation dynamics from RNA- and 4sU-seq time course experiments.** *Bioinformatics* 2015, **31**:2829-2835.
3. Graham B, Marçais A, Dharmalingam G, Carroll T, Kanellopoulou C, Graumann J, Nesterova TB, Bermange A, Brazauskas P, Xella B, et al: **MicroRNAs of the miR-290-295 Family Maintain Bivalency in Mouse Embryonic Stem Cells.** *Stem Cell Reports* 2016, **6**:635-642.
4. Nesterova TB, Popova BC, Cobb BS, Norton S, Senner CE, Tang YA, Spruce T, Rodriguez TA, Sado T, Merkschlager M, Brockdorff N: **Dicer regulates Xist promoter methylation in ES cells indirectly through transcriptional control of Dnmt3a.** *Epigenetics Chromatin* 2008, **1**:2.
5. Godoy PM, Barczak AJ, DeHoff P, Srinivasan S, Etheridge A, Galas D, Das S, Erle DJ, Laurent LC: **Comparison of Reproducibility, Accuracy, Sensitivity, and Specificity of miRNA Quantification Platforms.** *Cell Rep* 2019, **29**:4212-4222 e4215.
6. Linsen SE, de Wit E, Janssens G, Heater S, Chapman L, Parkin RK, Fritz B, Wyman SK, de Bruijn E, Voest EE, et al: **Limitations and possibilities of small RNA digital gene expression profiling.** *Nat Methods* 2009, **6**:474-476.
7. Eulalio A, Huntzinger E, Nishihara T, Rehwinkel J, Fauser M, Izaurralde E: **Deadenylation is a widespread effect of miRNA regulation.** *RNA* 2009, **15**:21-32.
8. Braun JE, Huntzinger E, Fauser M, Izaurralde E: **GW182 proteins directly recruit cytoplasmic deadenylase complexes to miRNA targets.** *Mol Cell* 2011, **44**:120-133.
9. Fabian MR, Cieplak MK, Frank F, Morita M, Green J, Srikumar T, Nagar B, Yamamoto T, Raught B, Duchaine TF, Sonenberg N: **miRNA-mediated deadenylation is orchestrated by GW182 through two conserved motifs that interact with CCR4-NOT.** *Nat Struct Mol Biol* 2011, **18**:1211-1217.
10. Tucker M, Valencia-Sanchez MA, Staples RR, Chen J, Denis CL, Parker R: **The transcription factor associated Ccr4 and Caf1 proteins are components of the major**

- cytoplasmic mRNA deadenylase in *Saccharomyces cerevisiae*. *Cell* 2001, **104**:377-386.
11. Denis CL, Chen J: **The CCR4-NOT complex plays diverse roles in mRNA metabolism.** *Prog Nucleic Acid Res Mol Biol* 2003, **73**:221-250.
 12. Li X, Pritykin Y, Concepcion CP, Lu Y, La Rocca G, Zhang M, King B, Cook PJ, Au YW, Popow O, et al: **High-Resolution In Vivo Identification of miRNA Targets by Halo-Enhanced Ago2 Pull-Down.** *Mol Cell* 2020, **79**:167-179 e111.
 13. Babiarz JE, Ruby JG, Wang Y, Bartel DP, Blelloch R: **Mouse ES cells express endogenous shRNAs, siRNAs, and other Microprocessor-independent, Dicer-dependent small RNAs.** *Genes Dev* 2008, **22**:2773-2785.
 14. Han YC, Vidigal JA, Mu P, Yao E, Singh I, Gonzalez AJ, Concepcion CP, Bonetti C, Ogradowski P, Carver B, et al: **An allelic series of miR-17 approximately 92-mutant mice uncovers functional specialization and cooperation among members of a microRNA polycistron.** *Nat Genet* 2015, **47**:766-775.
 15. Seitz H: **On the Number of Functional microRNA Targets.** *Mol Biol Evol* 2019, **36**:1596-1597.
 16. Wu L, Fan J, Belasco JG: **MicroRNAs direct rapid deadenylation of mRNA.** *Proc Natl Acad Sci U S A* 2006, **103**:4034-4039.
 17. Fabian MR, Mathonnet G, Sundermeier T, Mathys H, Zipprich JT, Svitkin YV, Rivas F, Jinek M, Wohlschlegel J, Doudna JA, et al: **Mammalian miRNA RISC recruits CAF1 and PABP to affect PABP-dependent deadenylation.** *Mol Cell* 2009, **35**:868-880.
 18. Wakiyama M, Takimoto K, Ohara O, Yokoyama S: **Let-7 microRNA-mediated mRNA deadenylation and translational repression in a mammalian cell-free system.** *Genes Dev* 2007, **21**:1857-1862.
 19. Helwak A, Kudla G, Dudnakova T, Tollervey D: **Mapping the human miRNA interactome by CLASH reveals frequent noncanonical binding.** *Cell* 2013, **153**:654-665.
 20. Guttman M, Russell P, Ingolia NT, Weissman JS, Lander ES: **Ribosome Profiling Provides Evidence that Large Noncoding RNAs Do Not Encode Proteins.** *Cell* 2013, **154**:240-251.
 21. Leung AK, Young AG, Bhutkar A, Zheng GX, Bosson AD, Nielsen CB, Sharp PA: **Genome-wide identification of Ago2 binding sites from mouse embryonic stem cells with and without mature microRNAs.** *Nat Struct Mol Biol* 2011, **18**:237-244.
 22. Wang Y, Baskerville S, Shenoy A, Babiarz JE, Baehner L, Blelloch R: **Embryonic stem cell-specific microRNAs regulate the G1-S transition and promote rapid proliferation.** *Nat Genet* 2008, **40**:1478-1483.
 23. Obad S, dos Santos CO, Petri A, Heidenblad M, Broom O, Ruse C, Fu C, Lindow M, Stenvang J, Straarup EM, et al: **Silencing of microRNA families by seed-targeting tiny LNAs.** *Nat Genet* 2011, **43**:371-378.
 24. Khan AA, Betel D, Miller ML, Sander C, Leslie CS, Marks DS: **Transfection of small RNAs globally perturbs gene regulation by endogenous microRNAs.** *Nat Biotechnol* 2009, **27**:549-555.

Dear Ana,

Thank you for submitting your revised manuscript, we have now received the reports from the three initial referees (see comments below). I am pleased to say that they overall find that their comments have been satisfactorily addressed and now support publication. Referee #2 however has remaining concerns regarding GW182 and the lncRNA reporter constructs. Please address these concerns by textual edits and additional discussion, and also provide a brief point-by-point response when submitting the revised manuscript. In addition, I would like to ask you to also address a number of editorial issues that are listed in detail below. Please make any changes to the manuscript text in the attached document only using the "track changes" option. Once these remaining issues are resolved, we will be happy to formally accept the manuscript for publication.

Referee #1:

In the revised version of their manuscript, the authors have addressed all points that I had raised on their previous version. They have added new experimental data and clarified statements that were unclear. I am satisfied and find that the authors adequately responded to my comments.

Referee #2:

The authors have made some changes to try to address my concerns. However, one of the major points (i.e. investigating if altering GW182 levels or determining whether GW182 RIP captures select lncRNAs) they were not able to address in spite of trying. Regarding another point, they originally used reporters with mutated putative miRNA binding sites on the lncRNA and attempted to show that the mutations relieved repression as compared to WT, which they could not show. In the revised manuscript they made some larger deletions instead of mutations and saw an effect, but they cannot conclude for sure which 'miRNA target sites' are actually involved for sure.

Thus, although the data are of value, they are mostly correlative. However, in light of the coronavirus pandemic, I recommend publishing the paper but with the provision that they should highlight the shortcomings noted above.

Referee #3:

The major points and concerns have been addressed in the revised version.

We would like to thank the reviewers for their supportive feedback and insightful suggestions that helped us shape and improve the manuscript.

Referee #1:

In the revised version of their manuscript, the authors have addressed all points that I had raised on their previous version. They have added new experimental data and clarified statements that were unclear. I am satisfied and find that the authors adequately responded to my comments.

Referee #2:

The authors have made some changes to try to address my concerns. However, one of the major points (i.e. investigating if altering GW182 levels or determining whether GW182 RIP captures select lncRNAs) they were not able to address in spite of trying.

We would like to thank the reviewer for the feedback. To the best of our knowledge GW182 RIP has only been carried out in human and in our hands the antibody used in this experiment does not work in mouse. The alternative to this experiment would be to endogenously tag GW182. We reasoned that the establishment and characterization of these cells entails lengthy experimental tests and goes beyond the scope of this work. Indeed, we use two independent genome-wide AGO2-RIP datasets to provide evidence that candidate lncRNA interacts with AGO2, which is known to be part of the same complex as GW182. Addition of GW182 would provide further support that lncRNAs interact with functional miRISC which is already supported by AGO2 binding and MRE validation experiments. We have edited our description of MREs in lncRNAs to ensure it accurately represents the data presented in the discussion (page 14, 4th paragraph continuing into page 15 1st paragraph).

Regarding another point, they originally used reporters with mutated putative miRNA binding sites on the lncRNA and attempted to show that the mutations relieved repression as compared to WT, which they could not show. In the revised manuscript they made some larger deletions instead of mutations and saw an effect, but they cannot conclude for sure which 'miRNA target sites' are actually involved for sure.

The reviewer misunderstood the difference between this and the previous version of the manuscript. As described in our previous response to the reviewer's comment 2 (that we copy below) we changed the way the data was represented to ensure it was clear that the changes we observed upon MRE mutation were not an indirect effect of *Dicer* loss of function. We conclude from this analysis that the mutated MREs contribute but that other mESC expressed miRNAs are likely to interact with this transcript.

Previous comment by the reviewer:

2. In the final figure, the authors fuse the lncRNA-c1 to the 3' of a GFP reporter and see an increase in its relative expression in DICER KO vs WT mES cells; however mutating the miRNA target sites does not seem to have a major impact on this reporter (Figure 4E, right column vs left column). Thus the effects on the construct is most likely indirect via DICER KO as opposed to directly targeting the lincRNA-c1 via

a number of miRNAs. In addition, it is unclear how the authors normalized for transfection efficiency of the construct.

Previous response to the reviewer's comment:

To account for differences in transfection efficiency between experiments and cell types, we normalize reporter expression (relative to *Actin-β* and *Polymeraseβ*) to the expression of Neomycin. Neomycin is transcribed from an independent promoter in the same vector. This is mentioned in the figure legends and detailed in the methods section.

“Gene expression levels relative to *Actin-β* and *Polymeraseβ* of transfected candidates were normalized to Neomycin expression to account for differences in transfection efficiency between different cell types and experiments”.

Whereas we agree with the reviewer that the MRE mutations do not fully rescue the effects of miRNAs on *GFP-IncRNA-c1* expression, we disagree that this is an indication that the observed differences are likely an indirect effect of *Dicer* loss of function. We attribute the reviewer's comment to the way the data was depicted in the previous version of the manuscript and that we have now modified.

Figure 4C- Expression of *GFP-IncRNA-c1* Δ MRE relative to *GFP-IncRNA-c1* (y-axis) in ethanol treated (WT, circles) or 4-OHT treated, miRNA-depleted cells (KO, triangles, x-axis). Four independent biological replicates were treated, transfected and analyzed by RTqPCR. For all RTqPCR analyses, transcript expression was first normalized by the amount of *Act-β* and *PolIII* and next by the total amount of transfected vectors per cell estimated based on the levels of relative *Neomycin* expression. Each point corresponds to the results of one independent biological replicate. Statistics: NS- p-value > 0.05, *-p-value < 0.05, **-p-value < 0.01 and ***-p-value < 0.001. Uncropped blots used for assembly of panels F and H are provided in Figure 4 Source Data.

As can be appreciated in the modified version Figure 4D, MRE deletion in *IncRNA-c1* is associated with increased abundance of reporter construct in WT cells relative to reporter gene with intact MREs. Importantly the increase we observe in *GFP-IncRNA-c1* Δ MRE expression relative to *GFP-IncRNA-c1* is significantly larger in WT cells. This observation supports that MRE deletion primarily affects the levels of reporter genes in WT cells, as expected. We attribute the incomplete rescue of MRE mutations, to the presence of functional binding sites for other miRNAs. As can be appreciated in Extended View File 8 we mutated 3 of the 17 MREs in *IncRNA-c1* for mESC expressed miRNAs, respectively. Whereas we chose to focus on MREs for the most highly expressed miRNAs in mESCs [13] a number of other less abundant

miRNAs in these cells have predicted MREs and can also impact the levels of targets. We modified the following section to clarify this point.

Thus, although the data are of value, they are mostly correlative. However, in light of the coronavirus pandemic, I recommend publishing the paper but with the provision that they should highlight the shortcomings noted above.

Referee #3:

The major points and concerns have been addressed in the revised version.

Thank you again for submitting the final revised version of your manuscript for our consideration. I am pleased to inform you that we have now accepted it for publication in The EMBO Journal.

Your article will be processed for publication in The EMBO Journal by EMBO Press and Wiley, who will contact you with further information regarding production/publication procedures and license requirements.

Corresponding Author Name: Ana Claudia Marques

Manuscript Number: EMBOJ-2020-104569R